# Multi-hierarchical profiling the structure-activity relationships of engineered nanomaterials at nano-bio interfaces

Xiaoming Cai[1], Jun Dong[2], Jing Liu[3], Huizhen Zheng[1], Chitrada Kaweeteerawat[4], Fangjun Wang[3], Zhaoxia Ji[5,6] & Ruibin Li [1]

Increasing concerns over the possible risks of nanotechnology necessitates breakthroughs in structure–activity relationship (SAR) analyses of engineered nanomaterials (ENMs) at nano-bio interfaces. However, current nano-SARs are often based on univariate assessments and fail to provide tiered views on ENM-induced bio-effects. Here we report a multi-hierarchical nano-SAR assessment for a representative ENM, $Fe_2O_3$, by metabolomics and proteomics analyses. The established nano-SAR profile allows the visualizing of the contributions of seven basic properties of $Fe_2O_3$ to its diverse bio-effects. For instance, although surface reactivity is responsible for $Fe_2O_3$-induced cell migration, the inflammatory effects of $Fe_2O_3$ are determined by aspect ratio (nanorods) or surface reactivity (nanoplates). These nano-SARs are examined in THP-1 cells and animal lungs, which allow us to decipher the detailed mechanisms including NLRP3 inflammasome pathway and monocyte chemoattractant protein-1-dependent signaling. This study provides more insights for nano-SARs, and may facilitate the tailored design of ENMs to render them desired bio-effects.

[1] State Key Laboratory of Radiation Medicine and Protection, Collaborative Innovation Center of Radiological Medicine of Jiangsu Higher Education Institutions, School of Public Health, School of Radiation Medicine and Protection, Soochow University, Suzhou, Jiangsu 215123, China. [2] Wuhan Academy of Agricultural Science, Wuhan, Hubei 430000, China. [3] CAS Key Laboratory of Separation Sciences for Analytical Chemistry, National Chromatographic R&A Center, Dalian Institute of Chemical Physics, Chinese Academy of Sciences (CAS), Dalian 116023, China. [4] National Nanotechnology Center (NANOTEC), National Science and Technology Development Agency (NSTDA), Klong Nueng 12120, Thailand. [5] California NanoSystems Institute, University of California, Los Angeles, CA 90095, USA. [6] Living Proof, Inc., Cambridge, MA 02142, United States. Correspondence and requests for materials should be addressed to Z.J. (email: zji@cnsi.ucla.edu) or to R.L. (email: liruibin@suda.edu.cn)

Physicochemical properties of engineered nanomaterials (ENMs) have been demonstrated to have decisive roles in nano-bio interactions[1]. Given the rapidly increasing number of ENMs as well as their diverse physicochemical properties including size, shape, surface area, surface reactivity, mechanical strength, etc.[2], the in vitro structure–activity relationship (SAR) studies on ENMs have significantly promoted the development of nanobiotechnology[3–5]. In general, nano-SAR analyses have enabled the determination of key physicochemical properties of ENMs that are responsible for evoking a target bio-effect in the organism[1,6], allowed bio-hazard ranking of various new ENMs[7], and facilitated the engineering design of biocompatible materials by tailored functionalization[8]. However, current nano-SAR analyses only focus on the influence of a single property (size, shape, or surface charge) of ENMs to a bio-effect (e.g., apoptosis, necrosis, autophagy, or inflammation)[2]. Considering some increasingly raised bottleneck problems in nanotechnology, such as various ENM-induced nanotoxicities[3,4], and severe clinical translation barriers in nanomedicine[9], there is a demand for tiered views of nano-SARs.

Omics is an attractive theme in biological science, aiming at system-level understanding of biological organisms. Several omics-based technologies including genomics, proteomics, and metabolomics have been developed for systematic analyses of biomolecules (nucleic acids, proteins, or metabolites) expressed in cells or tissues[10]. Recently, some progress has been made using omics to investigate protein corona on ENM surfaces[11], examine ENM-induced cell signaling changes[12,13], define the routes of ENM trafficking[14], and decipher cytotoxicity mechanisms[15]. A few attempts have been made to use single omics for nano-SAR assessments[16–18]. However, as proteins and metabolites are the executors or end products of signaling pathways and multi-omics analyses offer a better view of the global biological changes[19], we hypothesized that multi-hierarchical nano-SAR assessments could be achieved via coupling of proteomics and metabolomics analyses. As engineered iron oxide nanoparticles have been widely used in constructions[20], pigments[21], biomedicine[22,23], and its global production had reached to 1.83 billion in 2015, we decided to demonstrate our hypothesis using $Fe_2O_3$ nanoparticles in THP-1 cells, a macrophage-like cell line, which are the first port of entry for the ENMs exposed to mammalian systems[7,24].

In this study, we engineered a series of iron oxide nanoparticles to assess their SARs. The metabolomics and proteomics changes induced by $Fe_2O_3$ particles are examined in THP-1 cells. A multi-hierarchical nano-SAR profile is established by integration of the physicochemical properties of $Fe_2O_3$ particles, biological effects, and their correlation coefficients. The identified nano-SARs are selectively validated by deciphering the detailed mechanisms in vitro and in vivo.

## Results

### Preparation and characterization of $Fe_2O_3$ nanoparticles.
Given that various nanorods such as $CeO_2$, $AlOOH$, and lanthanide materials or nanoplates (e.g., Ag nanoplates) were demonstrated to be more reactive than other shapes[25–27], we synthesized a series of $Fe_2O_3$ nanoparticles with different morphologies and sizes, including four hexagonal nanoplates (P1~P4) with controlled diameters and thicknesses, and four nanorods (R1~R4) with systematically tuned lengths and diameters. Transmission electron microscopy (TEM) was used to determine the size and morphology of all $Fe_2O_3$ particles. Fig. 1a shows that the diameters of $Fe_2O_3$ nanoplates range from 45 to 173 nm and their thicknesses are 16~44 nm, whereas the lengths and diameters of nanorods are 88~322 and 20~53 nm, respectively. We further calculated the ratios of diameter to thickness

for the nanoplates and length to diameter for nanorods, respectively, and denoted them as aspect ratios (ARs). The ARs of $Fe_2O_3$ nanoplates and nanorods are 1.0~10.8 and 1.7~8.0, respectively. The surface areas were 16~27 $m^2$/g, determined by Brunauer–Emmett–Teller method (Table 1).

X-ray diffraction analysis (XRD) was performed to determine the crystal structure of $Fe_2O_3$. Supplementary Figure 1 shows the XRD patterns for selected nanoplates and nanorods, P2 and R2. All diffraction peaks could be indexed to rhombohedral $\alpha$-$Fe_2O_3$ phase (JCPDS no. 33-0664) and no other impurity peaks were detected.

The hydrodynamic sizes in RPMI medium were assessed by dynamic light scattering (DLS), showing ranges from 170 to 380 nm for plates and 420–540 nm for rods (Table 1). The surface charges of $Fe_2O_3$ particles were determined by a Zeta potential analyzer (ZPA). All particles showed very similar Zeta potential values of 0 to − 10 mV in cell culture media, which reflects the formation of protein corona on particle surfaces.

We used the 2',7'-dichlorofluorescein (DCF) assay to investigate the surface reactivity of $Fe_2O_3$ nanoparticles. The DCF assay is based on a mechanism that nonfluorescent 2',7'-dichlorodihydrofluorescein (DCFH) could be converted to the highly fluorescent DCF by oxidation. This assay has been widely used to assess the radicals or abiotic reactive oxygen species generation on nanoparticle surface[6,8]. $Co_3O_4$ nanoparticles have been demonstrated to exhibit high surface reactivity in the DCF assay and were used as a positive control[28]. As shown in Fig. 1b, c, $Fe_2O_3$ nanoplates are more reactive than nanorods and P3 exhibits the highest surface oxidative capability. These differences in surface reactivity may result from their crystal facets. XRD analysis shows that (104) is the dominant facet in $Fe_2O_3$ nanoplates as compared with (110), which is the strongest peak for the nanorods (Supplementary Figure 1). This is consistent with several earlier studies showing that (104), as the dominant facet of $\alpha$-$Fe_2O_3$ nanoplates, is highly catalytically active[29]. Therefore, the high surface reactivity of nanoplates observed here can be attributed to the exposure of more active (104) facets.

### Exploring $Fe_2O_3$ induced metabolite changes.
Metabolites are the end products of diverse intracellular processes, so the changes of cell metabolome can reflect cell responses to stimuli. We performed a global metabolomics study to explore the metabolic changes induced by $Fe_2O_3$ nanoparticles in THP-1 cells, a myeloid cell line that is often used as an in vitro model for studying the effects of engineered nanoparticles on immune cells[30]. As described in the experimental section, the metabolites in THP-1 cells after $Fe_2O_3$ treatment were extracted and separated by reversed-phase liquid chromatography (LC) for nontargeted mass spectrometry (MS) detection on a high-resolution time-of-flight (TOF) mass spectrometer in both positive (ESI + ) and negative (ESI −) ionization modes. Using the XCMS software, 8001 and 3498 features were obtained from the LC-MS data collected in ESI + and ESI − mode, respectively. One-way analysis of variance (ANOVA) was used to screen metabolite differences associated with $Fe_2O_3$ treatment. The significance of each feature was determined by its false discovery rate (FDR) truncated at 0.05. As a result, 1867 and 938 discriminating features were screened in the data of ESI + and ESI − mode, respectively. Fig. 2a shows a heat map of the significant features. Compared with the control, $Fe_2O_3$-treated samples show increases in most of the detected features. R4 induces the most significant metabolic changes in THP-1 cells; R1, R2, R3, and P3 show moderate effects, whereas P1, P2, and P4 are relatively bio-inert.

We further performed a hierarchical clustering analysis of metabolites to calculate the Euclidean distances (EDs) between

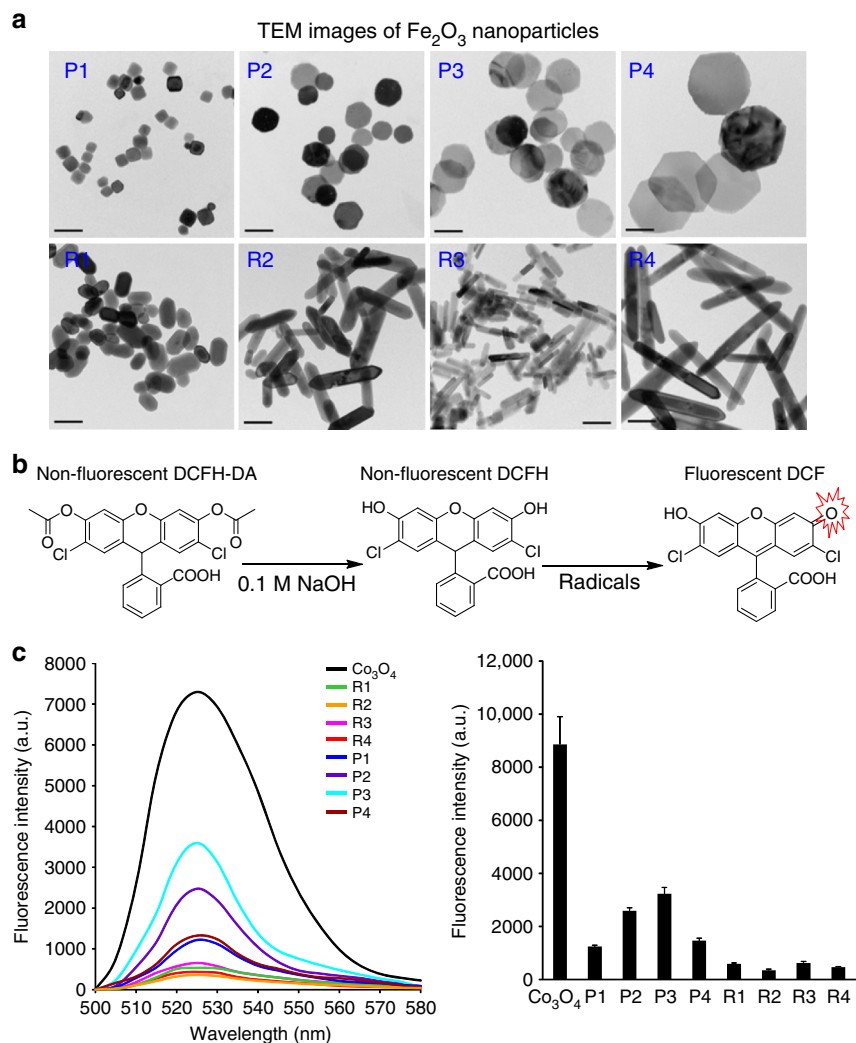

**Fig. 1** Characterization of $Fe_2O_3$ nanoparticles by TEM and DCF assay. **a** TEM images, **b** mechanism of DCF assay, and **c** surface reactivity of $Fe_2O_3$ nanoparticles. TEM samples were prepared by placing a drop of the particle suspensions (50 µg/mL in DI $H_2O$) on the grids. To assess the surface reactivity of $Fe_2O_3$ samples, 95 µL aliquots of 25 ng/mL DCFH were added into each well of a 96-multiwell black-bottom plate and mixed with 5 µL of nanoparticle suspensions at 5 mg/mL, followed by 2 h incubation. A SpectraMax M5 microplate reader was used to record the fluorescence emission spectra of DCF agent at an excitation wavelength of 490 nm. Error bar represents SD, $n = 3$. Scale bar in the TEM images is 100 nm

### Table 1 Quantitative characterization of $Fe_2O_3$ nanoparticles

| Properties | Detection methods | Quantitative characterization | | | | | | | |
|---|---|---|---|---|---|---|---|---|---|
| | | **Plates** | | | | **Rods** | | | |
| | | **P1** | **P2** | **P3** | **P4** | **R1** | **R2** | **R3** | **R4** |
| Length (nm) | TEM | NA | NA | NA | NA | 88 ± 8 | 181 ± 11 | 116 ± 12 | 322 ± 26 |
| Diameter (nm) | TEM | 45 ± 3 | 84 ± 5 | 122 ± 7 | 173 ± 6 | 53 ± 8 | 38 ± 5 | 20 ± 4 | 40 ± 7 |
| Thickness (nm) | TEM | 44 ± 7 | 23 ± 3 | 18 ± 2 | 16 ± 3 | NA | NA | NA | NA |
| Aspect ratio | Cal.* | 1.0 ± 0.1 | 3.7 ± 0.2 | 6.8 ± 0.1 | 10.8 ± 0.3 | 1.7 ± 0.2 | 4.5 ± 0.4 | 5.8 ± 0.5 | 8.0 ± 0.7 |
| Hydrodynamic size (nm) | DLS | 175 ± 10 | 246 ± 7 | 366 ± 16 | 378 ± 8 | 422 ± 12 | 463 ± 8 | 578 ± 8 | 536 ± 4 |
| Zeta potential (nm) | ZPA | − 7.9 ± 0.8 | − 3.6 ± 0.6 | − 5.1 ± 0.4 | − 3.7 ± 0.7 | − 6.5 ± 0.7 | − 10.7 ± 1.6 | − 8.5 ± 0.6 | − 5.2 ± 2.1 |
| Surface area ($m^2$/g) | BET | 22 ± 1.6 | 17.9 ± 1.0 | 17.4 ± 0.6 | 16.8 ± 0.4 | 18.3 ± 1.5 | 20.8 ± 1.8 | 26.9 ± 2.3 | 21.3 ± 1.8 |
| Surface activity ($10^3$ a.u.) | DCF | 1.25 ± 0.08 | 2.59 ± 0.11 | 3.23 ± 0.15 | 1.46 ± 0.08 | 0.59 ± 0.04 | 0.35 ± 0.03 | 0.63 ± 0.07 | 0.47 ± 0.03 |

*Calculation (Cal.): Aspect ratio $= \frac{\text{Length (or Diameter)}}{\text{Diameter (or Thinkness)}}$

the control and nanoparticle-treated samples[31]. This parameter was used to quantitatively describe the global metabolite profile changes induced by $Fe_2O_3$ and a longer distance usually means more disruptions to the homeostasis of cellular metabolism, which could be considered as a bio-activity index of stimuli at systemic levels[32]. As shown in Fig. 2a and Table 2, the ED

ranking of different $Fe_2O_3$ particles is R4 > R3 > R2 > P3 > R1 > P2 > P1 > P4, which is consistent with the observation on the metabolite heat map. Linear regression analysis was used to investigate the relationships between EDs and the physicochemical properties of $Fe_2O_3$ nanoparticles (Supplementary Figure 2). According to the $r^2$-values of regression models, surface reactivity

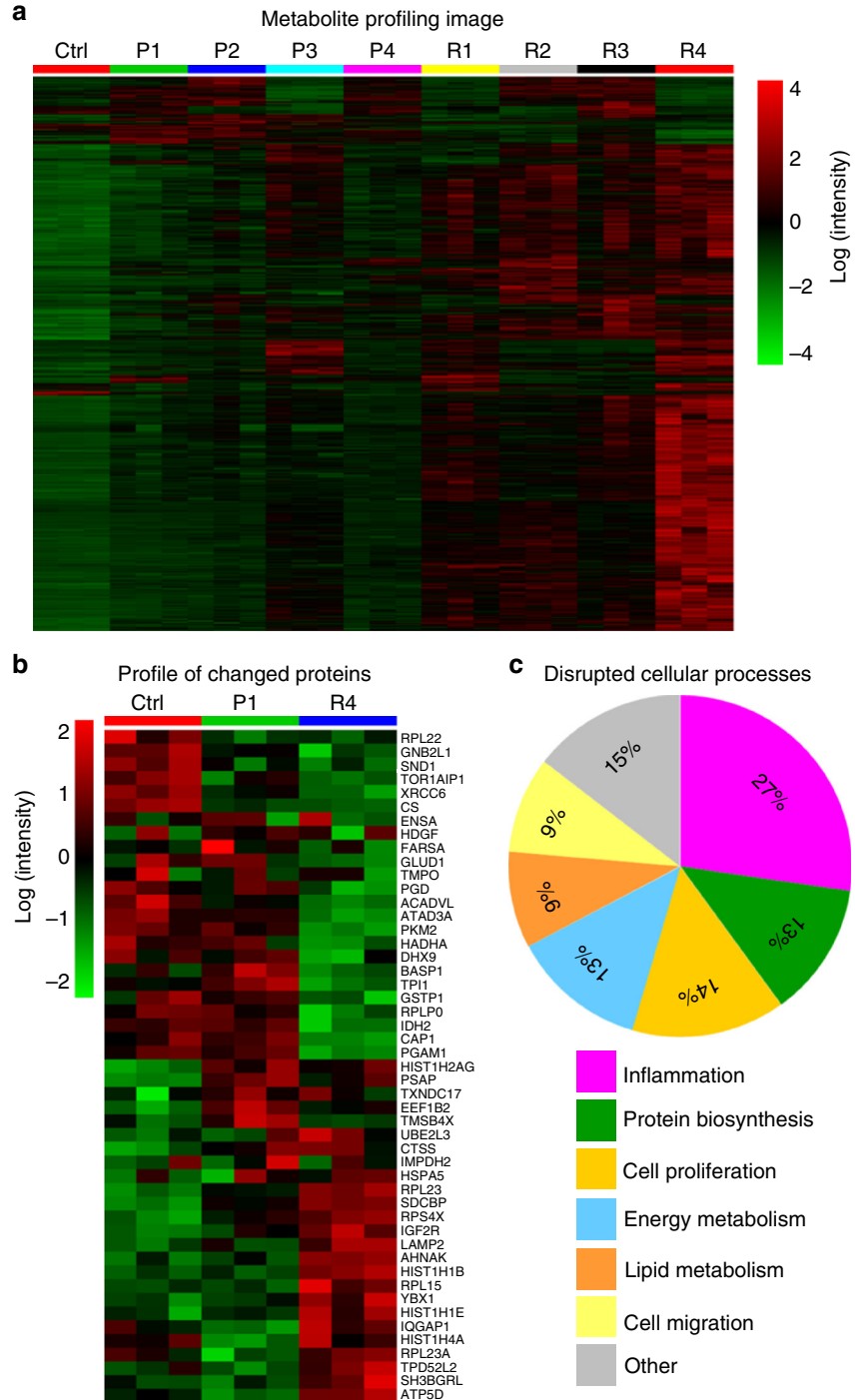

**Fig. 2** Fe$_2$O$_3$ induced metabolomics and proteomics changes and relationship. **a** Metabolite profile of THP-1 cells exposed to Fe$_2$O$_3$ library. After 24 h treatment, the cell samples ($n = 3$ independent experiments) were collected to extract proteins and metabolites. For metabolomics analysis, the log-transformed normalized peak intensities of metabolites in all the cell samples were expressed using red, black, or green colors in a heat map. **b** Fe$_2$O$_3$ nanoparticle-induced protein expression changes. For proteomics analysis, a heat map was plotted in a similar way as metabolites, showing 49 significantly changed proteins in proteomics analysis. **c** The pathways or bio-effects related with these proteins were determined by KEGG and UniprotKB database. The percentages of the differential proteins involved in each specific bio-effects of Fe$_2$O$_3$ were shown in the pie chart

and AR are the dominant physicochemical properties responsible for the global metabolite changes in THP-1 cells, which account for 91.25% and 99.44% of ED variations in nanoplates and nanorods, respectively (Supplementary Table 1).

Putative identification of the discriminating features was conducted using a PIUMet platform, which was developed for untargeted metabolomics by Pirhaji et al.[33]. Three hundred and

fourteen out of the 2805 identified features are matched to 417 potential metabolites in the HMDB database (Supplementary Data 1). As shown in Supplementary Figure 3, metabolic pathway analysis with MetaboAnalyst revealed that the putatively identified metabolites were responsible for 14 pathways including sphingolipid, tryptophan, phenylalanine, pyrimidine, glycerophospholipid, β-alanine, D-glutamine and D-glutamate, tyrosine,

purine, sulfur, glutathione and propanoate metabolisms, as well as primary bile acid, pantothenate and CoA biosynthesis (Supplementary Table 2).

**Discovery of the bio-effects of Fe$_2$O$_3$ particles by proteomics.** Proteins, as a major executor of signaling pathways in biological organisms, are involved in many cellular processes. The bio-effects of Fe$_2$O$_3$ nanoparticles in cells could be determined by identification of the proteome changes. As the metabolomics profile suggests that R4 and P1 are the most bioactive and bio-inert, respectively, these two materials were selected and exposed to THP-1 cells for proteomics analysis. The protein expression was analyzed by a nanoscale LC coupled to tandem MS (nano-LC-MS/MS) as described in the experimental section and 785 proteins were identified for statistical analysis.

To discover the biomarkers related to the bio-effects of Fe$_2$O$_3$, one-way ANOVA was performed. As a result, 49 identified proteins with $p$-values < 0.01 and FDRs < 0.05 were considered to be significantly changed after Fe$_2$O$_3$ treatment. The data were further integrated into a heat map, to visualize the expression levels of these proteins in control, P1 and R4 samples (Fig. 2b). Although R4 induced significant proteome changes in THP-1 cells including 25 upregulated and 24 downregulated proteins, P1 had negligible effects. We used Kyoto Encyclopedia of Genes and Genomes (KEGG) as well as UniprotKB database to investigate the impacts of the 49 proteins to cell pathways and functions. These proteins were found to mainly participate in six biological processes including inflammation, cell proliferation, energy metabolism, lipid metabolism, protein biosynthesis, and cell migration (Fig. 2c). Pearson's correlation analysis was used to explore the relationships between the changed proteins and the metabolites.

**Profiling the multi-hierarchical nano-SAR of Fe$_2$O$_3$ particles.** A heat map was plotted to quantitatively describe the influences of physicochemical properties on the bio-effects of Fe$_2$O$_3$ nanoparticles by regression analysis among the properties of nanoparticles, their metabolite changes, and bio-effects (Fig. 3). Although the zeta potential of Fe$_2$O$_3$ nanoplates has some effects on cell proliferations with coefficient $r^2$-value at 0.64, surface reactivity is the dominant property that impacts other five bio-effects as well as global cellular changes. For Fe$_2$O$_3$ nanorods, surface reactivity is responsible for the disruption of cell migration ($r^2 = 0.88$) and protein biosynthesis ($r^2 = 0.99$); particle length significantly affects the energy and lipid metabolism processes; AR has a major role in inflammation and cell proliferation. These results suggest that there is a different dominant property that best correlates with a specific bio-effect. This is the first time that the contributions of seven basic physicochemical properties of ENMs to their diverse bio-effects were analyzed together by plotting a nano-SAR profile at multi-hierarchical levels.

**Exploring the inflammatory effects of Fe$_2$O$_3$ particles.** The nano-SAR profile indicates that Fe$_2$O$_3$ nanoparticles may induce significant inflammatory effects in THP-1 cells. Among the changed proteins, 15 of them are involved in inflammation pathways, e.g., proactivator polypeptide (precursor of sphingolipid activator protein, PSAP), cathepsin S, and cation-independent mannose-6-phosphate receptor. These proteins have effects on phagocytosis and lysosomal dysfunction[34–36], implying a lysosome-involved mechanism. We further investigated this by detecting the pro-inflammatory cytokine release in THP-1 cells. Monosodium urate (MSU) was used as a positive control to evoke inflammatory response. Although Fe$_2$O$_3$ nanoparticles have little effect on cytokine production in THP-1 cells exposed to 0–100 μg/mL particles for 24 h (Supplementary Figure 4A), all Fe$_2$O$_3$-treated cells exhibit significant interleukin (IL)-1β and tumor necrosis factor (TNF)-α increase in dose-dependent manners at

**Table 2 EDs between Fe$_2$O$_3$-treated and control samples**

|  | P1 | P2 | P3 | P4 | R1 | R2 | R3 | R4 |
|---|---|---|---|---|---|---|---|---|
| Eulidean distance ($\times 10^5$) | 4.48 | 5.46 | 6.96 | 4.19 | 6.73 | 8.14 | 8.31 | 10.13 |

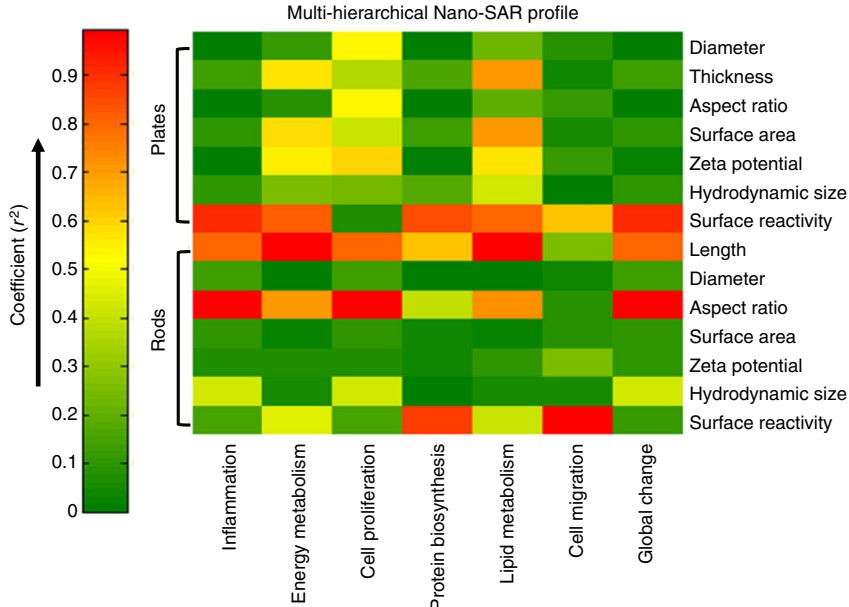

**Fig. 3** Multi-hierarchical profiling the nano-SAR of Fe$_2$O$_3$ particles. The relationships between the seven physicochemical properties of Fe$_2$O$_3$ nanoparticles and their bio-effects could be visualized by the heat map, which is established by a regression analysis among differential metabolites, differential proteins, and Fe$_2$O$_3$ properties

48 h (Fig. 4a, b and Supplementary Figure 4B). However, all these particles show little effects in cell viability (Supplementary Figure 5). At a 100 μg/mL exposure dose, R4 exhibits the highest inflammatory cytokine production; P3, R2, and R3 have moderate

effects, whereas P1, P2, P4, and R1 induce a small amount of cytokine release. This trend cannot be explained by the cellular uptake levels of $Fe_2O_3$ nanoparticles. $Fe_2O_3$ nanorods show lower dispersion stability in RPMI 1640 medium than nanoplates

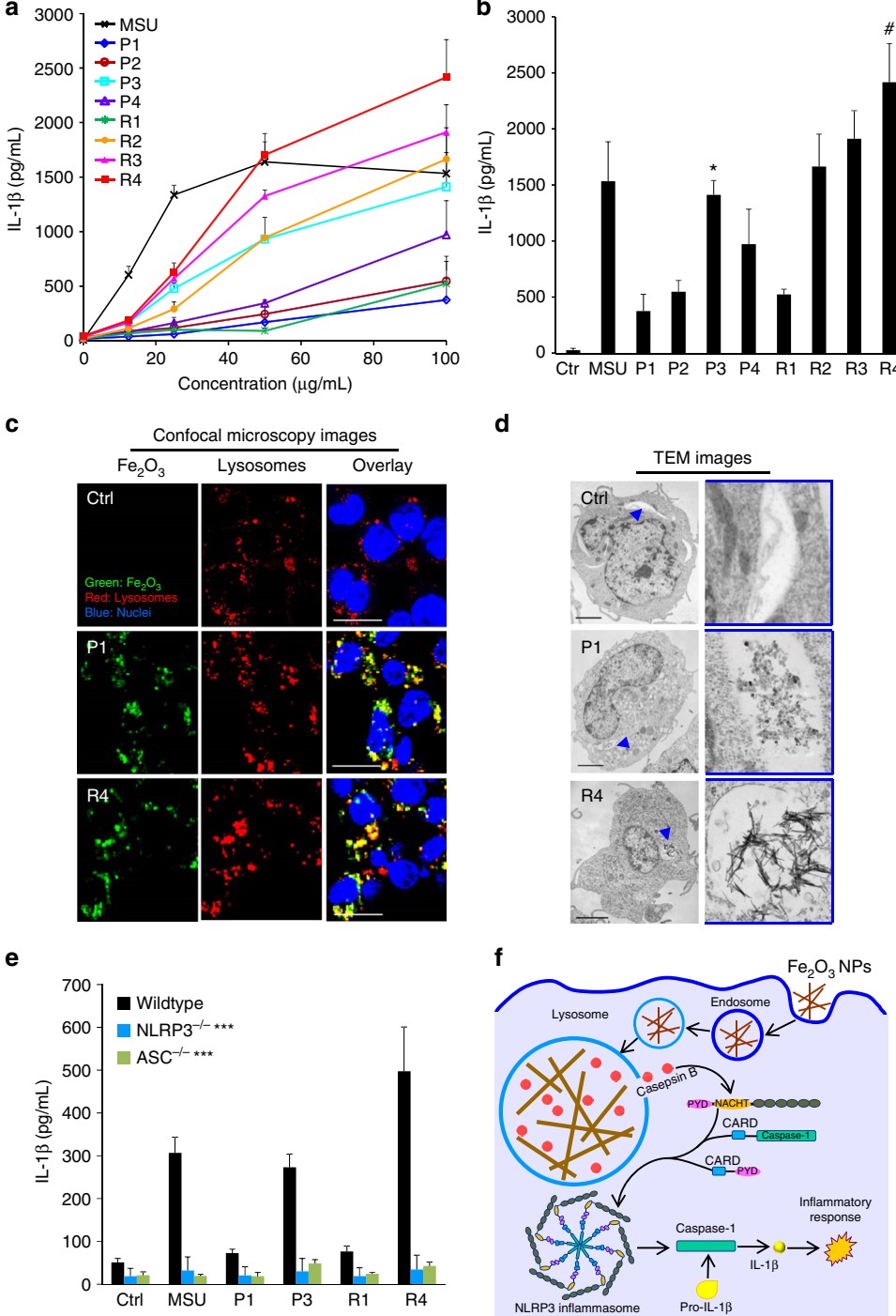

**Fig. 4** Determination of the inflammatory pathway of $Fe_2O_3$ in THP-1 cells. **a** Dose responses and **b** statistic comparisons of IL-1β in THP-1 cells exposed to $Fe_2O_3$ nanoparticles for 48 h. IL-1β production in THP-1 cells exposed to 0–100 μg/mL $Fe_2O_3$ nanoparticles for 48 h. *$p < 0.05$ compared with P1, P2, and P4; #$p < 0.05$ compared with other particles (two-tailed Student's $t$-test). After 48 h incubation of THP-1 cells with $Fe_2O_3$ particles, the supernatants were collected to quantify cytokine productions by ELISA. Error bar represents SD, $n = 3$. **c** Confocal microscopy and **d** TEM imaging of internalized $Fe_2O_3$ nanoparticles. THP-1 cells exposed to pristine or FITC-labeled $Fe_2O_3$ nanoparticles were washed, fixed, and stained for TEM or confocal microscopy imaging. Hoechst 33342 dye (blue) and Alexa Fluor 594-labeled antibodies (red) were used to identify nuclei and lysosomes, respectively. Scale bars for the TEM and confocal microscopy images are 5 μm and 20 μm, respectively. **e** Comparison of IL-1β production in wild-type, NLRP3$^{-/-}$, and ASC$^{-/-}$ THP-1 cells exposed to $Fe_2O_3$ nanoparticles, ***$p < 0.001$ compared with particle-treated wild-type cells (two-tailed Student's $t$-test). Data in the line and bar graphs are shown as mean ± SD from four independent replicates. **f** Schematic to explain the inflammatory mechanism

(Supplementary Figure 6), although there is no difference within nanorods or nanoplates. The results suggest that $Fe_2O_3$ nanorods are more bioavailable to cells with relatively higher cellular uptake than the nanoplates (Supplementary Figure 7). Consistent with the nano-SAR profile, cytokine productions by rods and plates have good correlations with their ARs and surface reactivity, respectively.

To understand the detailed mechanisms involved in $Fe_2O_3$-induced inflammatory effect, we further treated THP-1 cells with cytochalasin D, a cytoskeletal inhibitor of endocytosis, before exposure to the $Fe_2O_3$ particles. All cells treated with cytochalasin D showed a decrease in IL-1β release, suggesting that cellular uptake is essential in generating the inflammatory effect (Supplementary Figure 8). We examined the cellular uptake pathways of $Fe_2O_3$ nanoparticles by detecting the internalized nanoparticles in normal cultured (37 °C, without cytochalasin D treatment), 4 °C cultured, and cytochalasin D (10 μg/mL)-pretreated cells. After 2 h incubation with $Fe_2O_3$ nanoparticles, we used inductively coupled plasma-optical emission spectrometry to detect the iron contents in THP-1 cells. As shown in Supplementary Figure 9, all four materials show significantly decreased cellular internalization in 4 °C cultured cells and cytochalasin D-treated cells without inter-material differences, suggesting that both nanorods and nanoplates could be taken into cells by endocytosis, and the particle morphologies do not have an impact on their cellular uptake pathways. Confocal microscopy was used to study the cellular uptake of fluorescein isothiocyanate (FITC)-labeled $Fe_2O_3$ nanoparticles and we found that most of the labeled nanoparticles colocalized with an Alexa Fluor 594-labeled LAMP1-positive compartment with colocalization coefficients ranging from 73% to 93% by Image J analysis (Fig. 4c). This suggests that $Fe_2O_3$ nanoparticles were mainly taken into the lysosomes of THP-1 cells. TEM data confirmed that $Fe_2O_3$ nanoparticles are encapsulated into vesicular compartments of THP-1 cells (Fig. 4d). As lysosome is an acidifying environment, $Fe_2O_3$ particles tend to aggregate in this intracellular compartment and interact with its membranes. This likely has led to the lysosome membrane damage due to the reactive surfaces in nanoplates or geometric shape of $Fe_2O_3$ nanorods.

In order to explore the biological impact of $Fe_2O_3$ nanoparticles in lysosomal compartments, we asked whether that would impact lysosomal function. Confocal microscopy was used to study the subcellular localization of cathepsin B, a lysosomal enzyme capable of cleaving a Magic Red™-labeled substrate. As shown in Supplementary Figure 10A, untreated cells show a punctate distribution of Magic Red™, indicating that the enzyme is contained in intact lysosomes. However, after lysosomal damage by MSU, there is a diffuse cytosolic release of the fluorescence marker. Similarly, P3 and R4 nanoparticles induce cathepsin B release, whereas P1 and R1 nanoparticles are not associated with lysosomal damage. As cathepsin B is known to contribute to the activation of the NLRP3 inflammasome and IL-1β production[37], this may explain the severe inflammatory cytokine release in $Fe_2O_3$-treated THP-1 cells. The role of cathepsin B in NLRP3 inflammasome activation was further confirmed by using a cathepsin B inhibitor, CA-074-Me, which shows the inhibitory effect in IL-1β production (Supplementary Figure 10B). Moreover, we confirmed that active assembly of the NLRP3 inflammasome subunits is required for IL-1β production by using NLRP3- and ASC- gene knockdowns to show the interference in cytokine release in THP-1 cells (Fig. 4e). Based on the mechanism study, we deciphered the inflammatory pathway of $Fe_2O_3$ in THP-1 cells. As shown in Fig. 4f, $Fe_2O_3$ nanoparticles are internalized into lysosomes through endocytosis. Macrophage uptake and lysosomal processing of $Fe_2O_3$ nanoparticles further lead to the interaction with lysosome membrane. Because of the surface reactivity of $Fe_2O_3$ nanoplates and geometric shape of nanorods, these particles may induce lysosome damage, cathepsin B release into cytoplasm, recruitment of NLRP3, pro-caspase 1 and ASC subunits, NLRP3 inflammasome activation, and IL-1β release from the macrophages. IL-1β may further participate in a progressive march of inflammation events in organs.

**Examining the impacts of $Fe_2O_3$ particles on cell migration.** The nano-SAR profile also indicates that surface reactivity may be responsible for $Fe_2O_3$-induced cell migration. As monocyte chemoattractant protein-1 (MCP-1 or CC-chemokine ligand 2) is widely reported to be a critical factor for mediating arrest of the monocytic cells and directional migration[38,39], first we examined the effects of $Fe_2O_3$ particles on MCP-1 production. After 24 h exposure to $Fe_2O_3$ nanoparticles, significant MCP-1 production was detected in the supernatants of THP-1 cells and a higher level was observed for $Fe_2O_3$ plates than rods (Fig. 5a). Although P3 stimulated the highest MCP-1 secretion in THP-1 cells, P2 and P4 only had moderate effects. Then we transferred the supernatants of $Fe_2O_3$-treated THP-1 cells to the lower chambers of transwell systems to examine the effects in cell migration. As shown in Fig. 5b, $Fe_2O_3$ plates with high surface reactivity induce significant cell migration and P3 showing the highest level. To investigate whether the macrophage recruitment is a result of MCP-1 production, we examined the effects of supernatants from THP-1 cells exposed to bindarit and $Fe_2O_3$ particles on cell migration. Bindarit is a MCP-1 inhibitor and could effectively block all $Fe_2O_3$-induced MCP-1 productions. Results show that Bindarit treatment led to total reduction of cell migration, suggesting that $Fe_2O_3$ plates could induce MCP-1-dependent cell migration, and surface reactivity is the dominant property for this effect. Immune cell recruitment is an early statement in acute immune responses and involves transendothelial migration toward the stimulation site to protect health tissues[40]. Thus, $Fe_2O_3$ nanoplates display immunostimulatory functions and may serve as a modulator to activate immune cell recruitment, participating in monocyte or leukocyte migration.

**Validation of the inflammatory effects in mouse lungs.** In order to further confirm the nano-SAR of cellular inflammatory effects induced by $Fe_2O_3$ particles, we used an acute lung injury model to study the effect of oropharyngeal instilled nanoparticles in the whole lung. This study was performed with a particle dose of 2 mg/kg, which has been previously demonstrated to fall on the linear part of the dose–response curve for pulmonary exposure to metal oxide nanoparticles[28,41]. After 40 h exposure, the animals were killed to collect bronchoalveolar lavage fluid (BALF) and lung tissues. The cytokine release in BALF was determined by enzyme-linked immunosorbent assay (ELISA). As shown in Fig. 6a, most of the immune cells induced by $Fe_2O_3$ are neutrophils, which are dramatically boosted in P3 and R4 treated animal lungs. In addition, P3 and R4 induced significant release of cytokines including IL-1β (Fig. 6b), TNF-α (Fig. 6c), and LIX (Fig. 6d), which is consistent with their in vitro inflammatory responses. The migration effect of $Fe_2O_3$ particles was also validated by the MCP-1 production in BALF as well as hematoxylin and eosin staining of lung tissues. As shown in Fig. 6e, f, although P3 significantly elevates MCP-1 production and induces massive immune cell recruitment, P1 and R1 have little effect. Interestingly, R4 exhibits a limited effect in MCP-1 production but substantial immune cell recruitment in animal lungs, suggesting there may be other mechanisms involved in animal lungs. All these animal results demonstrated that the nano-SAR in $Fe_2O_3$-induced inflammatory and migration effects could be validated in vivo.

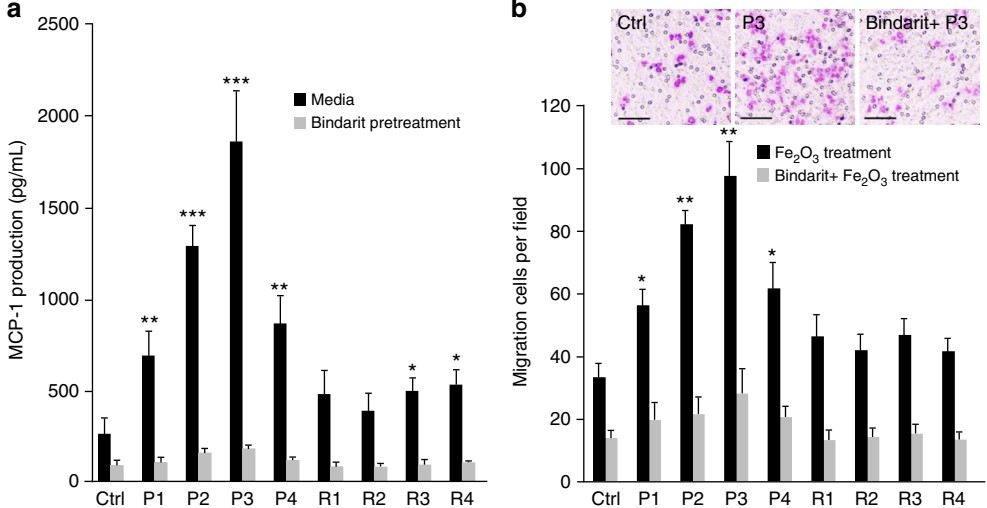

**Fig. 5** Effects of Fe$_2$O$_3$ nanoparticles on THP-1 cell migration. **a** MCP-1 production in the supernatants of Fe$_2$O$_3$-treated cells and **b** Transwell migration assays of THP-1 cells. THP-1 cells were pretreated with or without 100 µM bindarit for 3 h before exposure to 100 µg/mL Fe$_2$O$_3$ nanoparticles. After 24 h incubation, the supernatants of treated THP-1 cells were collected and divided into two portions. One was used for MCP-1 measurement by ELISA assay and the other one was added into the lower chambers of Transwell systems to incubate with THP-1 cells in upper chamber for 24 h. The migration cells were stained with Diff Quick agent and counted under microscope ($n = 5$ images for each treatment). *$p < 0.05$, **$p < 0.01$, ***$p < 0.001$ compared with ctrl (two-tailed Student's $t$-test). Scale bar is 50 µm. Error bar represents SD, $n = 3$

## Discussions

In this study, we established a multi-hierarchical nano-SAR assessment by simultaneously examining ENM-induced metabolite and protein changes in cells. Unlike traditional method that could only explore the nano-SAR between a specific physicochemical property of ENMs and individual bio-effects[2–4], a heat map was established in this study to assess the contributions of seven physicochemical properties of Fe$_2$O$_3$ particles to their six bio-effects in cells. The nano-SAR investigation could well facilitate the identification of key physicochemical properties that are responsible for ENM bio-effects, because the biological responses of ENMs have been demonstrated to result from their unique physicochemical properties[3,42].

To explore Fe$_2$O$_3$-induced bio-effects, metabolomics and proteomics analyses were performed in THP-1 cells exposed to Fe$_2$O$_3$ library. Among the 2805 significant metabolite peaks, 417 putative metabolites were identified from 314 features. Metabolic pathway analysis by MetaboAnalyst revealed that the putatively identified metabolites are responsible for 14 metabolic pathways (Supplementary Figure 3). These pathways were reported to have close relationships with Fe$_2$O$_3$-induced proteomics pathway changes (Supplementary Data 1). The identified metabolic pathways could also reflect the function changes of subcellular organelle including lysosome and mitochondria in Fe$_2$O$_3$-treated THP-1 cells. Among the 14 metabolic pathways, sphingolipids metabolism is significantly altered in the Fe$_2$O$_3$-treated samples ($p = 0.017$). Sphingolipids are bioactive lipids that are related to a diverse range of cellular responses including cell proliferation, autophagy, and inflammation[43]. The regulation mechanisms of various sphingolipids, such as ceramide[44], sphingosine-1-phosphate[45], ceramide-1-phosphate[46], and glycosphingolipids[47], in inflammatory processes have been extensively studied. Interestingly, although there are 2 ceramides, 2 sphingosine-1-phosphates, and 1 glucosylceramide in the list of 417 potential metabolites, proteomics only identified one significant changed protein, a single PSAP. This is a soluble membrane protein of lysosome and can be converted to sphingolipid activator protein, which is essential for the hydrolysis of sphingolipids in lysosomes[48]. Besides, sphingosine-1-phosphate

was demonstrated to have important roles in regulating cell migration via the G-protein-coupled receptors S1P[49]. Phosphatidic acid, an important member in glycerophospholipid metabolism, has been demonstrated to be a physiological regulator of ceramide-1-phosphate stimulated macrophage migration[50]. These metabolite pathways show correlations with the identified protein signaling changes and well support our findings on Fe$_2$O$_3$-induced NLRP3 inflammasome activation as well as MCP-1 dependent migration in THP-1 cells.

Recently, Zanganeh et al.[23] have demonstrated that iron oxide nanoparticle-induced pro-inflammatory macrophage polarization has an important role in tumor therapy. Syed et al.[51] discussed the nuclear factor-κB activation in ENM-induced inflammation, which dominates the production of some inflammatory cytokines including pro-IL-1β[52]. However, little is known on the nano-SARs involved in Fe$_2$O$_3$-induced IL-1β production as well as the detailed mechanism for the maturation of IL-1β. We answered these questions by setting up a combinatorial Fe$_2$O$_3$ library with precisely controlled size and shape, as well as a systematic assessment of their bio-effects. By regression analysis of particle properties, metabolite, and protein changes, AR and surface reactivity were identified as the key physicochemical properties responsible for the inflammatory effects of Fe$_2$O$_3$ nanorods and nanoplates, respectively. Fe$_2$O$_3$-induced cell migration was determined by surface reactivity. These findings were successfully validated in THP-1 cells and animal lungs. In vitro experiments further deciphered the MCP-1-dependent cell migration mechanism as well as the NLRP3 inflammatory pathway in Fe$_2$O$_3$-treated THP-1 cells.

Our study successfully achieved tiered view of the nano-SARs of Fe$_2$O$_3$ particles in THP-1 cells, providing more insights for tailored design of ENMs by modifying their physicochemical properties to acquire desired bio-effects. This allows us to reduce the toxicities of some hazardous ENMs[7,8,30] and to enhance the effects of nanomedicine[5,6,53]. The multi-hierarchical nano-SAR assessment method could also be potentially extended beyond Fe$_2$O$_3$ to other ENMs. This study has far-reaching implications for the sustainable development of nanotechnology.

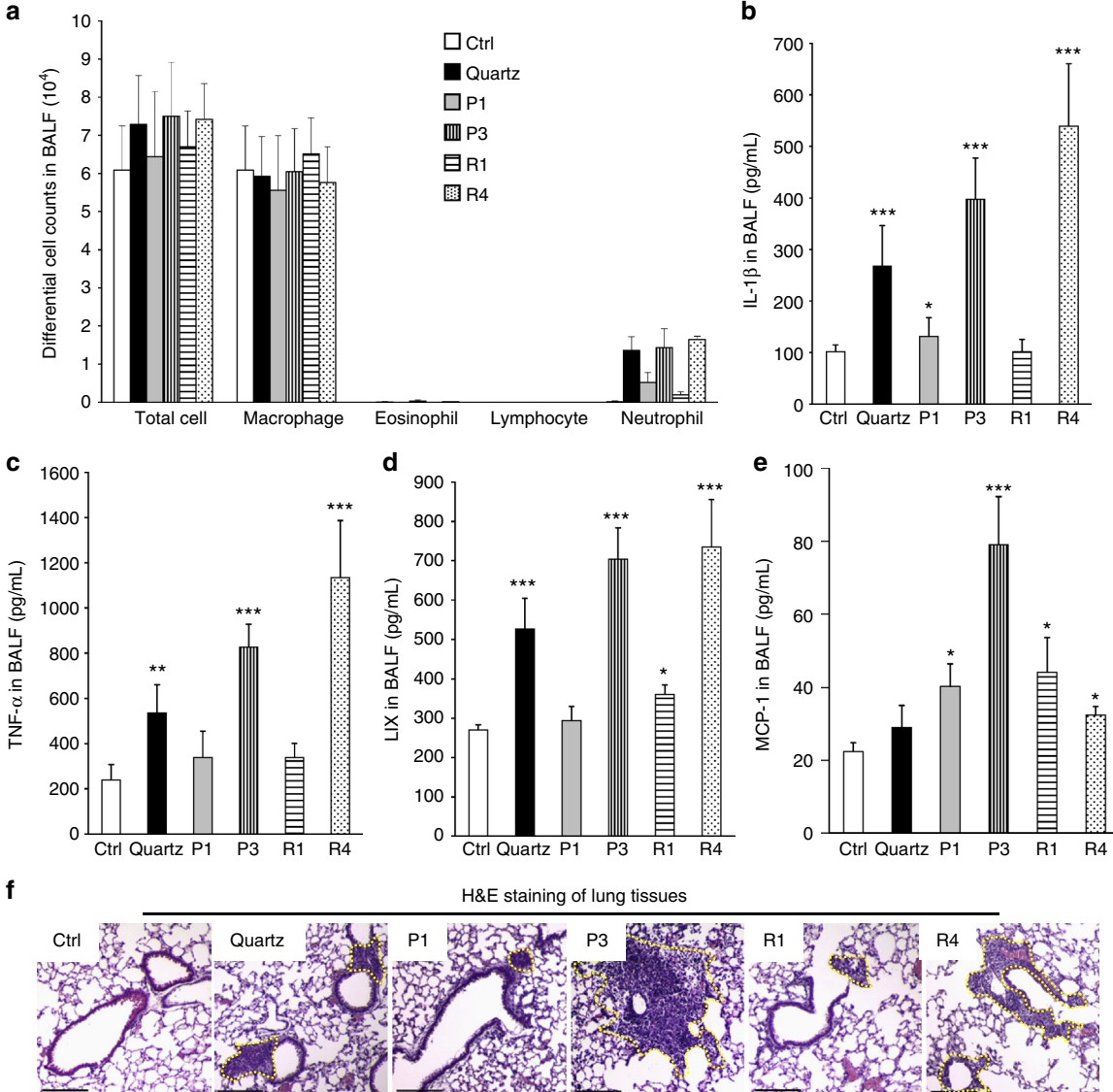

**Fig. 6** Inflammatory effects and cell migration in $Fe_2O_3$-treated animal lungs. **a** Differential cell counts, **b** IL-1β, **c** TNF-α, **d** LIX, and **e** MCP-1 productions in BALF, and **f** H&E staining of lung sections from $Fe_2O_3$-treated mice. Selected $Fe_2O_3$ nanoplates (P1, P3) and nanorods (R1, R4) were oropharyngeally administrated at 2 mg/kg ($n = 6$ mice in each group), whereas that animals received 5 mg/kg quartz exposure were used as positive control. After 40 h, animals were killed to collect BALF for differential cell counting as well as cytokine measurements, including IL-1β, LIX, TNF-α, and MCP-1. Data are shown as mean ± SD from four independent replicates. *$p < 0.05$, **$p < 0.01$, ***$p < 0.001$ compared with vehicle control (two-tailed Student's $t$-test). The dashed lines in H&E staining images show the immune cell recruitments and the scale bar represents 200 μm

## Methods

**Materials**. Magic Red™ Cathepsin B Assay Kit was purchased from Immunochemistry (Bloomington, MN, USA); Hoechst 33342, $H_2$-DCFDA, and Alexa Fluor 594-labeled goat anti-mouse IgG (Catalog Number R37121), were purchased from Life Technologies (Grand Island, NY, USA); ELISA kits for detection of murine or human IL-1β (Catalog Number 559603 or 557953), TNF-α (Catalog Number 560478 or 557953), and MCP-1 (Catalog Number 555260 or 555179) were purchased from BD biosciences (San Jose, CA, USA); LIX ELISA kit (Catalog Number DY443) was purchased from R&D (Minneapolis, MN, USA); MTS assay kit was purchased from Promega (Madison, WI, USA); anti-LAMP1 primary antibody (Catalog Number ab25630) and bindarit were purchased from Abcam (Cambridge, MA, USA); other chemicals, unless stated otherwise, were purchased from Sigma-Aldrich (St. Louis, MO, USA).

**In-house synthesis of $Fe_2O_3$ nanoparticles**. Both $Fe_2O_3$ nanorods and nanoplates were prepared using hydrothermal synthesis methods modified from published procedures[54,55]. In a typical $Fe_2O_3$ nanorod synthesis, a 10~16 mL of 0.86 M $FeCl_3 \cdot 6H_2O$ aqueous solution was prepared in a high-density polyethylene bottle, to which 4~10 mL of 1,2-propanediamine solution was added to form a 20 mL synthesis mixture. After stirring for 15 min, the synthesis mixture was transferred

to a 23 mL Teflon-lined stainless steel autoclave. The reaction was carried out in an electric oven at 180 °C under autogenous pressure and static conditions. After the reaction was complete, the autoclave was immediately cooled down in a water bath. The fresh precipitate was separated by centrifugation and washed with deionized water and ethanol alternatively for three cycles to remove ionic remnants. The final product was dried at 60 °C overnight under ambient environment. In a typical $Fe_2O_3$ nanoplate synthesis, 0.4325 g of $FeCl_3 \cdot 6H_2O$ was dissolved in 16 mL of ethanol with a small amount of water (1.1–4.0 mL) under vigorous stirring. After $FeCl_3 \cdot 6H_2O$ was completely dissolved, 1.3125 g of sodium acetate was then added and the resulting mixture was mixed for another 15 min. The reaction and final product collection were carried out the same way as those for the $Fe_2O_3$ naonorods.

**Preparation of $Fe_2O_3$ nanoparticles suspensions in media**. The $Fe_2O_3$ stock solutions were prepared by suspending particle powders in deionization (DI) $H_2O$ (5 mg/mL) and dispersed in a bath sonicator (Branson, Danbury, CT, USA, model 2510; 100 W output power; 42 kHz frequency) for 15 min. To prepare the desired concentrations of $Fe_2O_3$ suspensions, an appropriate amount of each $Fe_2O_3$ nanoparticle stock solution was added to cell culture media or PBS, and further dispersed using a sonication probe (Sonics & Materials, USA) at 32 W for 10 s before exposure to cultured cells or animals[56].

**Physicochemical characterization of Fe$_2$O$_3$ nanoparticles**. A particle suspension (50 μg/mL in DI H$_2$O) drop was added on the TEM grids for air-dry at room temperature. The TEM observation was performed on a JEOL 1200 EX instrument (accelerating voltage 80 kV). A Philips X'Pert Pro diffractometer equipped with CuKr radiation were used to obtain the XRD spectra. The hydrodynamic diameter and surface charge in water and cell culture media were measured by DLS coupled with ZPA (Brookhaven Instruments Corporation, Holtsville, NY, USA). DCF assay was used to evaluate the surface reactivity of Fe$_2$O$_3$ particles. In detail, 50 μg of H$_2$DCF-DA were mixed with 280 μL 0.01 M NaOH and incubated for 30 min at room temperature. The resulting solution was diluted with 1720 μL of a sodium phosphate buffer (25 mmol/L, pH = 7.4) to form 25 μg/mL DCF working solution. A 5 μL aliquots of nanoparticle suspension (5 mg/mL) were added into each well of a 96-multiwell black plate (Costar, Corning, NY) and then 95 μL amount of DCF working solution was added to each well, followed by 2 h incubation. DCF fluorescence emission spectra were recorded by a SpectraMax M5 microplate reader at an excitation wavelength of 490 nm[8].

**Cell culture and treatment**. THP-1 cells (TIB-202) were purchased from ATCC (Manassas, VA, USA) and cultured in RPMI 1640 supplemented with 10% fetal bovine serum. Before exposure to Fe$_2$O$_3$ nanoparticles, THP-1 cells were primed by 1 μg/mL phorbol 12-myristate acetate (PMA) overnight[7]. For cellular exposure, Fe$_2$O$_3$ nanoparticles were dispersed in complete RPMI 1640 medium at desired concentrations. Control sample was prepared by replacing the nanoparticle suspensions with pure water.

**Sample preparation for metabolomics and proteomics analyses**. THP-1 cells (1 × 10$^7$) were exposed to 100 μg/mL Fe$_2$O$_3$ nanoparticles for 24 h and cell samples including control and particle treatments were collected by centrifugation at 4 °C, 500 × g for 5 min. Each cell sample was equally divided into two portions. One portion is suspended in 1.5 mL cold lysis buffer (0.25 M sucrose, 50 mM Tris, 25 mM KCl, 5 mM MgCl$_2$, pH 7.4) for sonication on ice with 3 cycles of 10 s ON/OFF at a frequency of 20 kHz according to a reported method with a little modifications[57]. Methanol (6 mL) containing 0.5 mM L-Methionine-(methyl-13C,d3) and 1 mM D-Glucose-1,2,3,4,5,6,6-d7 as internal standards) pre-cooled at − 80 °C was added into the cell lysates and incubated 20 min at − 80 °C. The extraction mixture was centrifuged at 20,000 × g for 10 min at 4 °C to pellet the cell debris and the metabolite-containing supernatants were transferred to another 10 ml tube on dry ice. Aliquots of 500 μL extraction solution (− 80 °C) were added into each cell debris pellets, vortex for 1 min at 4 °C, and centrifuged to collect the supernatants and combined with the previous supernatants. The metabolite extracts were stored at − 80 °C after lyophilization. The other portion is lysed in 1 mL cold lysis buffer (50 mM Tris-HCl buffer, pH 7.4, 0.1% v/v Triton-100, 1 mM phenylmethylsulfonyl fluoride, 1% dithiothreitol, 8 M urea, 1 mM Na$_3$VO$_4$, 1 mM EDTA, 10 mM NaF, 10% protease inhibitor cocktail)[58] and probe sonicated on ice with 6 cycles of 10 s ON/OFF at a frequency of 20 kHz. After centrifugation at 20,000 × g for 10 min, each lysis supernatant was collected and added to 8 mL extraction solute (acetone/ethanol/acetic acid 50:50:0.1) pre-cooled at − 20 °C. Protein extracts were collected by centrifugation at 20,000 × g for 30 min after 24 h precipitation at − 20 °C. After lyophilization and re-dissolution in 8 M urea, the protein concentrations in each sample were determined by a Bradford method. The proteins were denaturalized, digested and desalted according to a reported method[58,59], to prepare peptide samples and stored at − 80 °C after lyophilization.

**Nontargeted metabolomics via LC-MS**. The metabolite pellets were re-suspended in 120 μL of buffer A (0.1% formic acid in 95:5 water/acetonitrile (ACN)) and 5 μL aliquots were injected for nontargeted LC-MS on a Shimadzu UFLC-XR system (Shimadzu Corporation, Japan) and an AB SCIEX TripleTOF 5600 + system (AB SCIEX, Foster City, CA). Samples were separated on a C18 reversed-phase high-performance liquid chromatography (HPLC) column (2.1 mm × 100 mm, 100 Å, 1.7 μm, Waters, Milford, MA) at 350 μL/min with a liner gradient of buffer A and buffer B (100% ACN) as follows: isocratic conditions at 100% A (0% B) for 2 min, a linear gradient from 100% A (0% B) to 30% (70% B) over 3 min, a linear gradient from 30% A (70% B) to 0% A (100% B) over 7 min, isocratic conditions at 100% B for 3 min. The column temperature was maintained at 50 °C.

To acquire the MS data, the MS conditions were set as follows: the ion spray voltage was + 5.5 kV (positive ion mode) or − 4.5 kV (negative ion mode); turbo spray temperature was 550 °C; nebulizer, heater, and curtain gases were at 50, 50, and 30 psi, respectively; TOF MS was scanned at the mass range of m/z 50~1200. Analyst v1.6.0 software (AB Sciex) was used to collected raw data, which was further converted into mzXML data format by proteoWizard software (Spielberg Family Center for Applied Proteomics, Los Angeles, CA) for further data processing.

The XCMS platform (https://metlin.scripps.edu/xcms/) was used for peak detection, retention time (RT) collection, and alignment. The parameters were set as follows: centWave settings for feature detection (maximal tolerated m/z deviation = 15 p.p.m., minimum peak width = 10 s, and maximum peak width = 80 s), obiwarp setting for RT correction (profStep = 1) and mzwid = 0.015, minfrac = 0.5, and bw = 5 for chromatogram alignment. The RTs, m/z values, and peak intensities of metabolites were exported to an Excel spreadsheet for

processing. Relative quantification of metabolite features were performed as following processes: (1) peak intensities were normalized to the internal standards: L-Methionine-(methyl-13C, d3) (m/z, 154.077) for positive mode, and D-Glucose-1,2,3,4,5,6,6-d7 (m/z, 186.099) for negative mode; (2) we compared the normalized peak intensities between control and particle-treated samples in Metaboanalyst (www.metaboanalyst.ca) to perform scatter plot, heat map, cluster, and ANOVA analysis. The parameters for data pro-processing in MetaboAnalyst were set as follows: removing features with > 90% missing values and estimating missing values using k-nearest neighbor method; using interquantile range method for data filtering to remove features from baseline noises; normalization by reference feature (154.077 for ESI + data and 186.099 for ESI − data), no data transformation, and pareto scaling were chosen for normalization procedure. For statistics, non-parametric ANOVA (Kruskal–Wallis Test) was performed and the adjusted p-value (FDR) cutoff is 0.05. A list of features with their p-values and FDR values was downloaded from the software. ED is used to measure the dissimilarity of samples with multivariate variables[31]. Here the ED between control and nanoparticle-treated samples is defined by the length between the two cluster centers. It was calculated by SPSS using following formula:

$$ED(C, T) = \sqrt{(t1 - c1)^2 + (t2 - c2)^2 + \ldots + (tn - cn)^2} \qquad (1)$$

where tn and cn are the log-transformed normalized peak intensities of metabolite n in nanoparticle-treated sample and control sample, respectively (n = 2805).

**Proteomics via nanoflow LC-MS/MS**. Lyophilized peptide samples were re-dissolved in 200 μL 0.1% formic acid for LC-MS/MS analysis. A LTQ Orbitrap Velos was equipped with an Accela 600 HPLC system (Thermo, San Jose, CA) to establish the nano-LC-MS/MS system. The nano-LC-MS/MS analysis was performed based on a previously reported method[58,59]. Briefly, the peptide samples were injected into a capillary trap column (200 μm i.d. × 4 cm, 120 Å, 5 μm), and then separated on an analytical column (15 cm × 75 μm i.d., 120 Å, 3 μm). Both of the columns were packed with C18 AQ beads. The separation buffer consisted of 0.1% (v/v) formic acid in DI H$_2$O (buffer A) and 0.1% (v/v) formic acid in ACN as buffer B, and the flow rate was 200 nL/min for HPLC-MS/MS analysis. The gradient from 5 to 40% (v/v) ACN was performed in 150 min. The MS and MS/MS spectra were collected by CID at 35% energy in a data-dependent mode with one MS scan followed by 20 MS/MS scans. The resolution was set at 60,000 for full MS and the scan range was set with m/z from 400 to 2000.

The resulted raw files were searched in MaxQuant (Version1.3.0.3) using Integrated Uniprot protein fasta database of human. Peptide searching was constrained using fully tryptic cleavage, allowing < 2 missed cleavage sites for tryptic digestion. Variable modifications included methionine oxidation (+ 15.9949 Da) and acetylation of protein N-term (+ 42.0106 Da), and static modification was set as cysteine carboxamidomethylation (+ 57.0215 Da). Precursor ion and fragment ion mass tolerances were set as 20 p.p.m. and 0.5 Da, respectively. The FDR for peptide and protein were < 1% and peptide identification required a minimum length of six amino acids.

Label-free relative quantification of the proteins was performed in this study. In detail, ion intensities for the peptides that constitute each protein were summed to generate a spectral intensity for the protein[60]. Summed intensities for the same protein obtained between samples were then used for one-way ANOVA analysis. The cell pathways and functions related to the identified differential proteins were explored by KEGG (http://www.kegg.jp/) and UniprotKB (http://www.uniprot.org/help/uniprotkb) database.

**IL-1β and TNF-α detection by ELISA**. IL-1β and TNF-α productions were detected in the culture media of THP-1 cells using human IL-1β and TNF-α ELISA Kit (BD; San Jose, CA, USA). Briefly, aliquots of 5 × 10$^4$ THP-1 cells were seeded in 0.1 mL complete medium and primed with 1 μg/mL PMA overnight in 96-well plates (Corning; Corning, NY, USA). Cells were treated with the desired concentration of the particle suspensions made up in complete RPMI 1640 medium, supplemented with 10% fetal bovine serum and 10 ng/mL lipopolysaccharide. The capture antibodies of IL-1β and TNF-α were added in 0.1 M sodium carbonate buffer (pH 9.5) with 250 × dilution for plate coating.

**Cell migration assay**. THP-1 cells were pretreated or not with 100 μM bindarit for 3 h before exposure to 100 μg/mL Fe$_2$O$_3$ nanoparticles. After 24 h incubation, the supernatants were collected for MCP-1 detection or cell migration assay, which was performed in 24-transwell plates with polycarbonate membranes of 8 μm pores (Corning, NY, USA). Lower wells were filled with 500 μL aliquots of the collected supernatants and 2 × 10$^5$ THP-1 cells (100 μL) were seeded into each of the upper wells. After incubation for 6 h, nonmigrated cells were scraped off from the upper side of the membrane and cells remaining within the pores or below the membranes were stained with Diff Quick[38]. Cell numbers were calculated under microscope by randomly selecting at least five individual fields for each sample.

**TEM imaging of Fe$_2$O$_3$ particles in THP-1 cells**. After exposure to 25 μg/mL Fe$_2$O$_3$ for 24 h, THP-1 cells were collected, washed, and fixed with 2% glutar-aldehyde in PBS. After 1 h post-fixation staining in 1% osmium tetroxide, a

dehydration process was performed by treating the cells in a graded series of ethanol, propylene oxide, and finally the cell pellets were embedded in Epon. A Reichert-Jung Ultracut E ultramicrotome was used to cut the TEM sections with ~50–70 nm thickness. The sections were further stained with uranyl acetate and Reynolds lead citrate before examining on TEM[53].

**Confocal microscopy imaging**. Leica confocal SP2 1P/FCS microscopes were used to visualize $Fe_2O_3$ uptake and cathepsin B release in THP-1 cells. High-magnification images were obtained under the ×63 objective. To visualize the cellular distribution, THP-1 cells were treated with 25 μg/mL FITC-labeled $Fe_2O_3$ nanoparticles for 6 h, fixed, and stained with Hoechst 33342, Anti-LAMP1 (250 × dilution), and Alexa Fluor 594-labeled antibodies (1000 × dilution) to visualize nuclei and lysosomes, respectively. For cathepsin B imaging, cells exposed to 100 μg/mL $Fe_2O_3$ particles for 16 h were stained with Magic Red™ Cathepsin B kit and Hoechst 33342 for confocal microscopy imaging.

**Inflammation test in mouse lungs**. Mice were exposed to nanoparticle suspensions using oropharyngeal aspiration at 2 mg/kg. Eight-week-old male C57Bl/6 mice purchased from Soochow University were used for animal experiments. All animals were housed under standard laboratory conditions that have been set up according to Soochow University guidelines for care and treatment of laboratory animals. These conditions were approved by the Chancellor's Animal Research Committee at Soochow University and include standard operating procedures for animal housing (filter-topped cages; room temperature at 23 ± 2 ℃; 60% relative humidity; 12 h light, 12 h dark cycle) and hygiene status (autoclaved food and acidified water). Animals were exposed by oropharyngeal aspiration[8]. Briefly, animals were anesthetized by intraperitoneal injection of ketamine (100 mg/kg)/xylazine (10 mg/kg) in a total volume of 100 μL. The anesthetized animals were held in a vertical position. Fifty-microliter aliquots of the nanoparticle suspensions in PBS were instilled at the back of the tongue to allow pulmonary aspiration of a dose of 2 mg/kg. Each experiment included control animals, which received the same volume of PBS. The positive control in each experiment received 5 mg/kg quartz. Each group included six mice. The mice were killed after 40 h exposure. BALF was collected by intra-trachea cannula syringing of 1 mL PBS, followed by formalin fixation to collect lung tissue. The BALF was used for performance of total and differential cell counts and measurement of IL-1β, TNF-α, MCP-1, and LIX levels. Lung tissue was stained with hematoxylin/eosin. Their capture antibodies were diluted in 0.1 M sodium carbonate buffer (pH 9.5) or PBS for plate coating according to the manufacturer's protocols.

**Statistical analysis**. All the experiments were repeated at least thrice with three to six replicates. Error bars represent the SD. All the cell and animal samples were randomly allocated into experimental groups by drawing lots. Results were expressed as mean ± SD of multiple determinations from at least three separate experiments. One-way ANOVA or Student's t-test was used for statistical analysis in MetaboAnalyst 3.0 or excel 2010. The difference is regarded statistically significant with $p \leq 0.01$ and $FDR \leq 0.05$ in omics analysis. Correlation analysis of the differential proteins and metabolites was performed in Matlab R2009b. The correlation is considered as statistically significant with correlation coefficient $(r) \leq -0.8$ or $r \geq 0.8$, and $p \leq 0.01$. The ED and linear regression analysis of the EDs and particle properties were achieved in SPSS 18.0. The profile of the SARs was also done in Matlab.

## Data availability

Metabolomics data have been deposited into MetaboLights with the code MTBLS721 and proteomics data have been deposited to data have been deposited to the ProteomeXchange Consortium via the PRIDE partner repository with the dataset identifier PXD010614. Other data that support the plots within this paper and other findings of this study are available in a generalist repository, Harvard Dataverse [https://dataverse.harvard.edu/privateurl.xhtml?token=6f339d25–7b45–48da-a2dd-5f733f8d69ea].

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

## Acknowledgements

This work was supported by the grants from the National Natural Science Foundation of China (Numbers 31671032 and 21804096), Key Project of Natural Science Foundation of the Higher Education Institutions of Jiangsu Province (Number 17KJA310003), Natural Science Foundation of Jiangsu Province (Number BK20180840), Suzhou Science and Technology Development Project (Number SYS201715), and a project funded by the Priority Academic Program Development of Jiangsu Higher Education Institutions (PAPD).

## Author contributions

X.C., Z.J. and R.L. conceived and designed the study. X.C. did most experiments. Z.J. synthesized the Fe2O3 nanoparticles and performed TEM, XRD, Zeta potential, and hydrodynamic size characterization with H.Z. J.D. performed the LC-MS analysis for metabolomics study. F.W. and J.L. contributed to the proteomics analysis. The writing of the paper was led by X.C. and R.L with participation from Z.J. and C.K.

## Additional information

**Competing interests:** The authors declare no competing interests.

