## [Peer Review File · Nature Communications]

Reviewers' comments:

Reviewer #1 (Remarks to the Author):

This manuscript developed multi-hierarchical nano-SAR assessment for Fe₂O₃ nanoparticles by metabolomics and proteomics analyses. They used Fe₂O₃ nanoparticles with different aspect ratio (nanorods) or surface reactivity (nanoplates). It is quite interesting to find that surface reactivity is responsible for Fe₂O₃-induced cell migration, the inflammatory effects of Fe₂O₃ are determined by aspect ratio (nanorods) or surface reactivity (nanoplates). They also discovered the detailed cellular mechanisms including NLRP3 inflammasome pathway and monocyte chemoattractant protein-1 dependent signaling. Overall, the experiments were well designed. y. The findings are useful. There are some concerns as below,

- 1) metabolomics and proteomics analyses can provide many detailed information, how to ensure the data quality is very important. So the procedure for the detailed sample collection and preparation, pretreatment and store of samples and measurement should be carefully provided.
- 2) Cellular uptake is essential in generating the inflammatory effect, I suggest to move Figure S6 to the main text. However, the pathway of internalization of nanoparticles is missing. The size and shape may have some impacts on uptake pathway.
- 3) The stability of Fe₂O₃ nanoparticles in biological environment is not known, in particular, in culture medium and mice.
- 4) How is the dispersity of Fe₂O₃ nanoparticles?
The English expression needs to be improved.

Reviewer #2 (Remarks to the Author):

General comments:

1. This manuscript reports on the application of proteomic and metabolomic analysis followed by bioinformatic data evaluation to study the biological effects of iron oxide nanomaterials of different size and shape using cell culture and in vivo model systems. Based on the increasing application of nanoscaled materials in everyday life, the study of the biological risks of nanomaterials is a currently very modern topic also of high scientific significance.
2. In the literature, it has been numerously demonstrated that unbiased, wholistic "omics" approaches are very well suited to generate biological hypotheses with respect to the biological effects evoked by nanomaterials. Moreover, it is very well appreciated that combinations of omics analysis are providing an appropriate means to filter the plethora of data obtained for the major biological mechanisms behind the interaction with nanomaterials and also derive biomarkers or pathways indicative of toxic effects of the materials under study. Moreover, wholistic omics approaches as described in this manuscript have numerously been utilized to study biological effects of other stimuli such as drugs or chemicals in general.
3. In due consequence, this referee believes that the approach proposed is not really innovative in terms of technology applied or workflow employed to transform the analytical data into biological knowledge. I do highly appreciate the very interesting findings about the mechanism of action of different morphologies and concentrations of iron oxide but such information should be more suited for a Journal devoted to nanomaterials or nanotoxicity. Also the connection between physico-chemical properties of the materials and biological effects are meriting publication.
4. Several claims in the manuscripts are clear overstatements and are either already well known facts or clear overstatements that are not sufficiently supported by the data presented, e. g.:
Page 3, lines 69 and 84: "However, current nano-SAR analyses only focus on the influence of a single property" and "However, so far, no attempts have been made for nano-SAR assessments".
A search in Google Scholar for "nanomaterial properties biological effects" yields 169.000 hits, for instance Kristin L. Aillon, Yumei Xie, Nashwa El-Gendy, Cory J. Berkland, M. Laird Forrest, Effects of nanomaterial physicochemical properties on in vivo toxicity, Advanced Drug Delivery

Reviews, Volume 61, Issue 6, 2009, Pages 457-466.

Page 15, line 340: "Based on the mechanism study, we for the first time deciphered the inflammatory pathway of Fe₂O₃ in THP-1 cells.", oxidative stress and inflammation have been very well described as a biological effects of nanomaterial exposure, see, e. g. Mo-Tao Zhu, Bing Wang, Yun Wang, Lan Yuan, Hua-Jian Wang, Meng Wang, Hong Ouyang, Zhi-Fang Chai, Wei-Yue Feng, Yu-Liang Zhao, Endothelial dysfunction and inflammation induced by iron oxide nanoparticle exposure: Risk factors for early atherosclerosis, Toxicology Letters, Volume 203, Issue 2, 2011, Pages 162-171.

Page 15, line 354: "we first examined the effects of Fe₂O₃ particles on MCP-1 production." MPP-1 action has also been described in the reference Zhu above.

Page 18, line 425: In this study, we pioneered a multi-hierarchical nano-SAR assessment by simultaneously examining ENM-induced metabolite and protein changes in cells. This statement is not at all supported by the data and conclusions presented in the manuscript, see points 5 and 6 below.

Page 19, line 437: "however, the nano-SAR information as well as the detailed inflammatory mechanism shrouds in mystery."; This is contradicted by numerous publications, such as Syed, S., Zubair, A. & Frieri, M. Immune Response to Nanomaterials: Implications for Medicine and Literature Review, Curr Allergy Asthma Rep (2013) 13: 50. <https://doi.org/10.1007/s11882-012-0302-3>.

5. I do see a significant problem with the combination of proteomics and metabolomics data. The number of 3400 to 8000 detected features is far too high to represent real metabolites detectable in a simple HPLC-MS analysis. In our experience, the majority of detected features is also present in the blank, so rigorous elimination of such false positives needs to be performed and documented. Moreover, no attempt was made to identify the "discriminating [metabolite] features", so I cannot see any possibility to link the features with any biological function. In due consequence, the linking of metabolome and proteome data does not make any biological sense, since the metabolite features are only m/z numbers that cannot be linked to metabolic pathways. Finally, the number of detected proteins is also very moderate since a HPLC-LTQ Orbitrap Velos setup would be able to detect more than 3000 proteins in a cell lysate.

6. The release of pro-inflammatory cytokines is also a well-known biological effect of exposure to ENM, see Mahmoud Elsabahy and Karen L. Wooley Cytokines as biomarkers of nanoparticle immunotoxicity, Chem.Soc.Rev., 2013, 42, 5552.

7. In conclusion, I do see a significant number of conceptual and technological deficiencies in this manuscript such that I believe it is not suitable for publication in Nature Communications.

Specific comments

- 1) Introduction: contains too many etc.: either name what is meant or leave it out
 - 2) Page 3, line 77: systemS biology is the commonly used term
 - 3) Page 3, line 80: genes are not biomolecules, correct would be nucleic acids
 - 4) Page 4, line 115: Brunauer-Emmet-Teller (BET) method (https://en.wikipedia.org/wiki/BET_theory)
 - 5) Page 7, line 165: ... metabolites ... were subjected to c18 reversed-phase column -> ... metabolites were separated by reversed-phase HPLC ...
 - 6) Page 7, line 168: high-resolution time-of-flight mass spectrometer
 - 7) Figure 2 c: This looks very fancy but does not really give a lot of useful information. As stated above, the linking of identified proteins to unidentified metabolite m/z values does not make any biological sense. From the figure I learn that there may be a positive, negative or null correlation between protein expression and metabolite concentration, which is trivial. Such an investigation only makes sense when the correlating metabolites and the metabolic pathways behind are identified.
 - 8) Figure 3: 3D bar graphs should be generally avoided; effects would be much better visible in a heat map, showing the correlation between ENM properties and biological effects.
 - 9) Figure 4: NALP3 and NLRP3 protein/gene names are used promiscuously, should be unified.
 - 10) Methods descriptions are insufficient or erroneous in many respects:
- a) Page 22, line 529: I do not believe that method is correctly described. Upon extraction of

lysates with methanol, metabolites should be in the supernatant and proteins/DNA in the pellet!

b) Relative protein and metabolite quantification and statistics are not sufficiently described. Also details about protein identification are missing.

c) The authors utilize a typical microscale ultrahigh pressure HPLC system to run a nanoscale HPLC column (I suppose 75 MICRO m instead of 75 MILLI m i.d.), How was that accomplished? For the HPLC method the authors refer to another published paper from the M. Mann group (ref. 43), but in the reference no methodological details are given but reference is made to another publication.

d) Data availability: availability from the corresponding author upon reasonable?? Request is not meeting modern standards. Data must be freely available in public data repositories.

We thank reviewers for their valuable comments on our manuscript (NCOMMS-18-04537) entitled: “Multi-hierarchical Profiling the Structure-Activity Relationships of Engineered Nanomaterials at Nano-Bio Interfaces”. Below please find detailed point-to-point responses to the reviewer’s comments and suggestions. A marked-up copy of the manuscript named “marked manuscript for review” was also uploaded, in which all the changes are in red. The major changes made since the original submission include:

1. We added more experimental and data processing descriptions for metabolomics and proteomics analysis.
2. We attempted to identify the putative metabolites and the results are shown in the Supplementary Excel Data.
3. We performed a metabolite pathway analysis based on the putatively identified metabolites and the result is shown in Figure S3.
4. We measured the UV-vis absorbance of Fe₂O₃ nanoparticles dispersed in cell culture media to evaluate their dispersion stability. The results are shown in Figure S6.
5. We compared the cellular uptake levels of Fe₂O₃ particles in THP-1 cells at normal culture condition (37 °C), 4 °C culture and with endocytosis inhibitor pretreatment to determine the cellular uptake pathways. The results are shown in Figure S9 in the revised manuscript.
6. Metabolomics and proteomics data have been deposited into MetaboLights (<https://www.ebi.ac.uk/metabolights/MTBLS721>; user name: xiaomingcai1982@hotmail.com; password: MTBLS721) and PRIDE (code: PXD010614; username: reviewer38153@ebi.ac.uk ; password: wzJEYRQP), respectively. Other data are available in a generalist repository, Harvard Dataverse (<https://dataverse.harvard.edu/privateurl.xhtml?token=6f339d25-7b45-48da-a2dd-5f733f8d69ea>).

Reviewer #1 (Remarks to the Author):

This manuscript developed multi-hierarchical nano-SAR assessment for Fe₂O₃ nanoparticles by metabolomics and proteomics analyses. They used Fe₂O₃ nanoparticles with different aspect ratio (nanorods) or surface reactivity (nanoplates). It is quite interesting to find that surface reactivity is responsible for Fe₂O₃-induced cell migration; the inflammatory effects of Fe₂O₃ are determined by aspect ratio (nanorods) or surface reactivity (nanoplates). They also discovered the detailed cellular mechanisms including NLRP3 inflammasome pathway and monocyte chemoattractant protein-1 dependent signaling. Overall, the experiments were well designed. The findings are useful.

Response: We thank the reviewer for considering that our experiments were well designed and the findings are useful.

There are some concerns as below,

- 1) Metabolomics and proteomics analyses can provide many detailed information,

how to ensure the data quality is very important. So the procedure for the detailed sample collection and preparation, pretreatment and store of samples and measurement should be carefully provided.

Response: We added a detailed description on proteomics and metabolomics sample preparation in experimental section in lines 574 to 599, as “THP-1 cells (1×10^7) were exposed to 100 $\mu\text{g}/\text{mL}$ Fe_2O_3 nanoparticles for 24 h, and cell samples including control and particle treatments were harvested by centrifugation at 4 $^\circ\text{C}$, 500 g for 5 min. Each cell sample was equally divided into two portions. One portion is suspended in 1.5 mL cold lysis buffer (0.25 M sucrose, 50 mM Tris, 25 mM KCl, 5 mM MgCl_2 , pH 7.4) for sonication on ice with 3 cycles of 10 sec ON/ OFF at a frequency of 20 kHz according to a reported method with a little modifications⁵⁷. Methanol (6 mL) containing 0.5 mM L-Methionine-(methyl- ^{13}C ,d3) and 1mM D-Glucose-1,2,3,4,5,6,6-d7 as internal standards) pre-cooled at -80 $^\circ\text{C}$ was added into the cell lysates and incubated 20 min at -80 $^\circ\text{C}$. The extraction mixture was centrifuged at 20000 g for 10 min at 4 $^\circ\text{C}$ to pellet the cell debris, and the metabolite-containing supernatants were transferred to a new 10 ml tube on dry ice. Aliquots of 500 μL extraction solution (-80 $^\circ\text{C}$) were added into each cell debris pellets, vortex for 1 min at 4 $^\circ\text{C}$ and centrifuged to collect the supernatants and combined with the previous supernatants. The metabolite extracts were stored at -80 $^\circ\text{C}$ after lyophilization. The other portion is lysed in 1 mL cold lysis buffer (50 mM Tris-HCl buffer, pH 7.4, 0.1% v/v Triton-100, 1 mM phenylmethylsulfonyl fluoride, 1% dithiothreitol, 8 M urea, 1 mM Na_3VO_4 , 1 mM EDTA, 10 mM NaF, 10% protease inhibitor cocktail)⁵⁸, and probe sonicated on ice with 6 cycles of 10 sec ON/ OFF at a frequency of 20 kHz. After centrifugation at 20000 g for 10 min, each lysis supernatant was collected and added to 8 mL extraction solute (acetone/ethanol/acetic acid 50:50:0.1) pre-cooled at -20 $^\circ\text{C}$. Protein extracts were collected by centrifugation at 20 000 g for 30 min after 24 h precipitation at -20 $^\circ\text{C}$. After lyophilization and re-dissolution in 8 M urea, the protein concentrations in each sample were determined by a Bradford method. The proteins were denaturalized, digested and desalted according to a reported method^{58,59}, to prepare peptide samples and stored at -80 $^\circ\text{C}$ after lyophilization. ”

2) Cellular uptake is essential in generating the inflammatory effect, I suggest to move Figure S6 to the main text. However, the pathway of internalization of nanoparticles is missing. The size and shape may have some impacts on uptake pathway.

Response: To address the reviewer’s concern, we compared the cellular uptake pathways of R1, R4, P1 and P3. We examined the uptake pathways of Fe_2O_3 nanoparticles by detecting the internalized nanoparticles in normal cultured (37 $^\circ\text{C}$, without cytochalasin D treatment), 4 $^\circ\text{C}$ cultured and cytochalasin D (10 $\mu\text{g}/\text{mL}$) pretreated cells. After 2 h incubation with Fe_2O_3 nanoparticles, we used ICP-OES to determine the iron contents in THP-1 cells. As shown in Figure R1, all four particles show significantly decreased cellular internalization in 4 $^\circ\text{C}$ cultured cells and cytochalasin D treated cells without inter-material differences, suggesting that both nanorods and nanoplates could be taken into cells by endocytosis, and the particle

morphologies do not impact their cellular uptake pathways. This figure has been added in revised manuscript as Figure S9, and its description was added in lines 300 to 308, as “We examined the cellular uptake pathways of Fe₂O₃ nanoparticles by detecting the internalized nanoparticles in normal cultured (37 °C, without cytochalasin D treatment), 4 °C cultured and cytochalasin D (10 µg/mL) pretreated cells. After 2 h incubation with Fe₂O₃ nanoparticles, we used ICP-OES to detect the iron contents in THP-1 cells. As shown in Figure S9, all four materials show significantly decreased cellular internalization in 4 °C cultured cells and cytochalasin D treated cells without inter-material differences, suggesting that both nanorods and nanoplates could be taken into cells by endocytosis, and the particle morphologies do not impact their cellular uptake pathways.”

Figure R1 Detection of internalized Fe₂O₃ in THP-1 cells at different culture conditions.

Four Fe₂O₃ nanoparticles including P1, P3, R1 and R4 were exposed to THP-1 cells cultured at 4 °C, 37 °C or cytochalasin D pretreated (10 µg/mL, 3 h) THP-1 cells. After 4 h incubation, the cells were rinsed with cold PBS and collected by centrifugation for cell digestion. The iron contents in cell lysates were measured by ICP-OES.

3) The stability of Fe₂O₃ nanoparticles in biological environment is not known, in particular, in culture medium and mice.

Response: We used UV-Vis spectroscopy to measure the absorbance of Fe₂O₃ dispersed in cell culture media at different time points. The suspension stability index

could be calculated by a previously reported formula: $\frac{A_0 - A_i}{A_0} \times 100\%$, where A₀ is the

initial absorbance of Fe₂O₃ suspensions, while A_i represents the absorbance at various timepoints (Wang et al, ACS Nano, 2010, 7241; Li et al, ACS Nano, 2013, 2352). As shown in Figure R2, the Fe₂O₃ nanoplates displayed the higher dispersion stability than Fe₂O₃ nanorods, with ca. 55% remaining suspended after 48 h. However, there are no significant differences between different nanorods or nanoplates. All considered, these results suggest that the relatively higher sedimentation rate of Fe₂O₃

nanorods would make them more bioavailable to cells settling at the bottom of the wells, leading to higher cellular uptake as shown in Figure R1. This new result has been added as Figure S6 in the revised manuscript.

We selected P3 and R4 as representative particles to evaluate their solubility in cell culture media and mouse lungs. As shown in Figure R3, P3 and R4 are very stable in cell culture media and more than 95% nanoparticles remained in culture media after 21 d incubation. P3 and R4 could be gradually eliminated from animal lung after receiving Fe₂O₃ exposure by oropharyngeal aspiration. At 21 d post exposure, there were only 25% P3 and 41% R4 present in lung tissue.

Figure R2 Dispersion stability of Fe₂O₃ nanoparticles in cell culture media

Fe₂O₃ nanoparticles were dispersed in RPMI 1640 media at 25 µg/mL by probe sonication. The absorbance at 550 nm was measured at 0, 0.5, 1, 2, 4, 6, 12, 24 and 48 h using a UV-Vis spectrometer. The suspension stability index could be calculated by a previously reported formula: $\frac{A_0 - A_i}{A_0} \times 100\%$, where A₀ is the initial absorbance of Fe₂O₃ suspensions and A_i represents the absorbance at various timepoints.

Figure R3 Stability of Fe₂O₃ nanoparticles in cell culture media and animal lungs

To determine the stability of Fe₂O₃ nanoparticles in cell culture media, P3 or R4 suspensions (25 µg/mL) in RPMI 1640 media were incubated at 37 °C. After 2 h, 40 h, 7 d or 21 d incubation, an aliquot of each Fe₂O₃ suspension was taken and the supernatant after centrifugation was collected to examine particle dissolution by ICP-OES. The percentages of particles present in suspensions reflect the stability of Fe₂O₃ nanoparticles in cell culture media. For the stability test in animal lungs, P3 or R4 were exposed to mouse lungs by oropharyngeal aspiration (2 mg/Kg) for 2 h, 40 h, 7 d or 21 d. Then the lung tissues were collected and digested to determine residual particles.

4) How is the dispersity of Fe₂O₃ nanoparticles?

Response: To answer this question, we compared the primary and hydrodynamic sizes of Fe₂O₃ nanoparticles in PBS and RPMI 1640 media. As shown in Table R1, both nanoplates and nanorods show agglomeration in PBS and cell culture medium compared to their primary sizes. However, particles show better dispersion in complete RPMI 1640 medium with fetal bovine serum. The presence of protein in cell media could significantly improve the dispersion of Fe₂O₃ nanoparticles due to the formation of protein corona.

Table R1 Primary and hydrodynamic sizes of Fe₂O₃ nanoparticles

Particles	Primary sizes (nm)	Hydrodynamic sizes (nm)	
		PBS	c-RPMI 1640
P1	45 × 44	213	175
P2	84 × 23	408	246
P3	122 × 18	489	366
P4	173 × 16	568	378
R1	88 × 53	536	422
R2	181 × 38	568	463
R3	116 × 20	661	578
R4	322 × 40	784	536

The English expression needs to be improved.

Response: We have improved English writing in revised manuscript.

Reviewer #2 (Remarks to the Author):

General comments:

1. This manuscript reports on the application of proteomic and metabolomic analysis followed by bioinformatic data evaluation to study the biological effects of iron oxide nanomaterials of different size and shape using cell culture and in vivo model systems. Based on the increasing application of nanoscaled materials in everyday life, the study of the biological risks of nanomaterials is a currently very modern topic also of high scientific significance.

Response: We thank the reviewer for considering our study as “a currently very modern topic also of high scientific significance”.

2. In the literature, it has been numerously demonstrated that unbiased, wholistic “omics” approaches are very well suited to generate biological hypotheses with respect to the biological effects evoked by nanomaterials. Moreover, it is very well appreciated that combinations of omics analysis are providing an appropriate means to filter the plethora of data obtained for the major biological mechanisms behind the interaction with nanomaterials and also derive biomarkers or pathways indicative of toxic effects of the materials under study. Moreover, wholistic omics approaches as described in this manuscript have numerously been utilized to study biological effects of other stimuli such as drugs or chemicals in general.

Response: We thank the reviewer to point out that “omics” approaches are very well suited to investigate the biological effects evoked by nanomaterials.

3. In due consequence, this referee believes that the approach proposed is not really innovative in terms of technology applied or workflow employed to transform the analytical data into biological knowledge. I do highly appreciate the very interesting findings about the mechanism of action of different morphologies and concentrations of iron oxide but such information should be more suited for a Journal devoted to nanomaterials or nanotoxicity. Also the connection between physico-chemical properties of the materials and biological effects are meriting publication.

Response: We thank the reviewer to point out that the mechanisms of action of different morphologies are very interesting findings and the connection between physico-chemical properties of materials and biological effects are meriting publication.

We do recognize that the omics techniques used in this study are not new. However, we have to point out that our study is aimed to make new findings on nano-SARs using combined omics, rather than develop new omics-based methods. Based on the proteomics and metabolomics analysis, we made three innovative findings in this

study: i) we achieved multi-hierarchical profiling of nano-SAR, which could simultaneously assess the contributions of seven physicochemical properties of Fe₂O₃ to six specific bio-effects; ii) based on the nano-SAR profile of Fe₂O₃, the surface reactivity of Fe₂O₃ plates and aspect ratio of Fe₂O₃ rods were found to be the key physicochemical properties that dominate the inflammatory or migration effects; iii) we deciphered the cascaded signaling events in Fe₂O₃-induced NLRP3 inflammasome activation and discovered the MCP-1 dependent immune cell recruitment by Fe₂O₃.

We think our study well fits the scope of Nature Communications because it's a multidisciplinary research including structure-activity relationships (nanoscience), NLRP3 and MCP-1 pathways (biology), lung inflammation (pathology) as well as omics (analytical chemistry). In addition, we made new and interesting findings on nano-SARs as well as cell pathways of Fe₂O₃-induced bio-effects. We believe that these innovations will arouse broad interest to the readers of Nature Communications.

4. Several claims in the manuscripts are clear overstatements and are either already well known facts or clear overstatements that are not sufficiently supported by the data presented, e. g.:

Page 3, lines 69 and 84: "However, current nano-SAR analyses only focus on the influence of a single property" and "However, so far, no attempts have been made for nano-SAR assessments". A search in Google Scholar for "nanomaterial properties biological effects" yields 169.000 hits, for instance Kristin L. Aillon, Yumei Xie, Nashwa El-Gendy, Cory J. Berkland, M. Laird Forrest, Effects of nanomaterial physicochemical properties on in vivo toxicity, *Advanced Drug Delivery Reviews*, Volume 61, Issue 6, 2009, Pages 457-466.

Response: We thank the reviewer for the comments. To address his/her concern, we have performed further analysis on the published literatures and made appropriate changes on our statement.

We typed the reviewer-suggested key words in Web of Science with 537 hits, and 156 papers mentioned the structure activity relationships (SARs). Among these publications, none of them simultaneously examined the influences of multiple physicochemical properties on one specific bio-effect, which supports our first statement, "However, current nano-SAR analyses only focus on the influence of a single property". To make this statement more clear, we made a little change on it, as "However, current nano-SAR analyses only focus on the influence of a single property (size, shape, or surface charge) of ENMs to individual bio-effect (e.g. apoptosis, necrosis, autophagy or inflammation)". In terms of the second statement, we made a substantial publication researches. Table R2 listed three most relevant publications (Kinaret et al, *ACS Nano*, 2017, 3786; Betrand et al, *Nature communications*, 2017, 777; He et al, *Nature communications*, 2018, 2393). Although omics technologies were used in all these studies, there are three major differences between our study and other three publications: i) Fe₂O₃ was investigated in our study;

ii) we made two new findings on nano-SARs as well as cellular mechanisms (see the second paragraph of our responses to comment 3); iii) we used combined omics rather than single omics approach. Based on this comparison, we agree with the reviewer that our second statement may be not appropriated. We made a little changes on our second statement as, “In addition, a few attempts have been made to use single omics for nano-SAR assessments¹⁶⁻¹⁸”.

We carefully examined the review paper (Advanced Drug Delivery Reviews, Volume 61, Issue 6, 2009, Pages 457-466) mentioned by the reviewer. This paper briefly summarized some nanotoxicity mechanisms (ROS generation, DNA damage, NF-κB activation, cell-cycle arrest, mutagenesis, and apoptosis), and discussed the influences of individual physicochemical properties including size, chemical composition and stability on nanotoxicity. However, four interesting findings in our study are not mentioned in this review paper: i) simultaneously profiling the contributions of seven physicochemical properties of Fe₂O₃ to six specific bio-effects; ii) the cascaded signaling events in Fe₂O₃-induced NLRP3 inflammasome activation; iii) Fe₂O₃-induced MCP-1 dependent immune cell recruitment; iv) the surface activity or aspect ratio of Fe₂O₃ particles determines their inflammatory or migration effects.

Table R2 Comparison with three most relevant papers

	ENMs	Omic Technique	In vitro model	Number of studied properties	Findings on Nano-SARs	Cellular findings	mechanism	Animal results
Our study	Fe ₂ O ₃	Proteomics, Metabolomics	THP-1 cell	Seven	Simultaneously profiling the impacts of seven properties to six bio-effects; Aspect ratio and surface reactivity are responsible for the inflammatory or migration effects	NLRP3 pathway dependent cell migration	inflammasome and MCP-1	Pulmonary inflammation
Kinaret et al' paper in ACS Nano	CNTs, Fulleren, Graphite Baytube	Transcriptomics	THP-1 cell	Four	Impacts of a single property (aspect ratio, diameter, length or surface area) to gene expressions	NA		Gene expression in mouse lungs
Bertrand et al' paper in Nat Commun	Polymer	Proteomics	Plasma	Two	Influences of sizes or PEG densities to corona formation	Low-density-lipoprotein receptor clearance of nanoparticles in vivo	determines the	Blood circulation time
He et al' paper in Nat Commun	Nanohorn, Nanotube	Proteomics	J774A.1 cell	One	Influences of shape on cell death	Weaker interaction leads to lower degree of cascade actions in cell toxicity	nano-GPNMB	NA

Page 15, line 340: “Based on the mechanism study, we for the first time deciphered

the inflammatory pathway of Fe₂O₃ in THP-1 cells.”, oxidative stress and inflammation have been very well described as a biological effects of nanomaterial exposure, see, e. g. Mo-Tao Zhu, Bing Wang, Yun Wang, Lan Yuan, Hua-Jian Wang, Meng Wang, Hong Ouyang, Zhi-Fang Chai, Wei-Yue Feng, Yu-Liang Zhao, Endothelial dysfunction and inflammation induced by iron oxide nanoparticle exposure: Risk factors for early atherosclerosis, Toxicology Letters, Volume 203, Issue 2, 2011, Pages 162-171.

Response: We agree with the reviewer that oxidative stress has been very well described as a biological effect of nanomaterial exposure. In our paper, we do not study this bio-effect. We respectfully disagree with the reviewer’s comment that inflammation has been very well described. We carefully examined the publication suggested by the reviewer. Table R3 showed the comparison of our and Zhu et al’ studies. There are 11 differences. The major findings on Fe₂O₃ in our paper have not been reported before (Figure R4). Zhu et al’ study was cited in the revised manuscript as ref 39.

Table R3 Comparison of our and Zhu et al’ studies

	Our study	Zhu et al’ study
Particle composition and shape	Fe ₂ O ₃ nanoplates or nanorods	Spherical Fe ₂ O ₃ and Fe ₃ O ₄
Particle number	Eight	Two
Characterized properties	Seven	Three
Cell model	U937, HAECs	THP-1
Cytokines	IL-1β	IL-8, ICAM-1
Validated bio-effects	Inflammatory and migration	Inflammation, ROS, cell death
Inflammatory pathway	NLRP3 inflammasome activation	NA
Toxicity-related properties	Aspect ratio and surface reactivity	composition
Nano-SAR	Simultaneously visualizing the impacts of seven properties to six bio-effects	Influences of composition on cytokine release, cell viability, ROS generation.
Animal model	Lung exposure	NA
In vivo findings	Pulmonary inflammation and immune cell recruitment	NA

Search History:

Set	Results	
# 2	0	ts=Fe2O3 and ts=nano* and ts=cell migration and ts=MCP-1 Timespan=All years Search language=Auto
# 1	0	ts=Fe2O3 and ts=nano* and ts=NLRP3 Timespan=All years Search language=Auto

Figure R4 Search summary of our two findings in Web of Science

We claimed two important findings on Fe₂O₃ in our manuscript, i.e., NLRP3 inflammasome activation and MCP-1 dependent cell migration. None of these effects have been reported.

Page 15, line 354: “we first examined the effects of Fe₂O₃ particles on MCP-1 production.” MPP-1 action has also been described in the reference Zhu above.

Response: We apologize for the confusion. We did not mean to claim that we were the first group to study the effects of Fe₂O₃ particles on MCP-1 production. To make this statement more clear, we rephrased it in the revised manuscript as “**First of all, we examined the effects of Fe₂O₃ particles on MCP-1 production**”.

Page 18, line 425: In this study, we pioneered a multi-hierarchical nano-SAR assessment by simultaneously examining ENM-induced metabolite and protein changes in cells. This statement is not at all supported by the data and conclusions presented in the manuscript, see points 5 and 6 below.

Response: We have made changes in the revised manuscript and provided point-by-point responses to Points 5 and 6. We hope the changes as well as our responses have addressed the reviewer’s concern.

Page 19, line 437: “however, the nano-SAR information as well as the detailed inflammatory mechanism shrouds in mystery.”; This is contradicted by numerous publications, such as Syed, S., Zubair, A. & Frieri, M. Immune Response to Nanomaterials: Implications for Medicine and Literature Review, *Curr Allergy Asthma Rep* (2013) 13: 50. <https://doi.org/10.1007/s11882-012-0302-3>.

Response: We carefully examined the publication by Syed et al. In this review paper, they discussed the NF-κB pathway in ENM-induced inflammation, which is responsible for the production of pro-IL-1β. This is not surprising because most stimuli have similar pro-IL-1β production pathway, a.k.a. Toll-like-receptor and NF-κB activation (Emma et al, *Trends in Immunology*, 2006, 352-357; Poeck et al, *Nature Immunology*, 2010, 63–69; Church et al, *Nature clinical practical practice rheumatology*, 2008, 34-42). However, different ENMs may lead to the maturation of pro-IL-1β via distinct mechanisms, including NLRP1, NLRP3, NLRC4, AIM2 and RIG-I inflammasome activation. Currently, it’s unclear whether Fe₂O₃ may lead to inflammasome activation (Figure R4). Therefore, we claimed, “however, the nano-SAR information as well as the detailed inflammatory mechanism shrouds in mystery.” To make this claim more accurate, we made a little change on it, as “**However, we know little on the nano-SARs involved in Fe₂O₃-induced IL-1β production as well as the detailed mechanism for the maturation of IL-1β.**” Syed et al’ study was cited as ref 51.

5. I do see a significant problem with the combination of proteomics and metabolomics data. The number of 3400 to 8000 detected features is far too high to represent real metabolites detectable in a simple HPLC-MS analysis. In our experience, the majority of detected features is also present in the blank, so rigorous elimination of such false positives needs to be performed and documented. Moreover, no attempt was made to identify the “discriminating [metabolite] features”, so I cannot see any possibility to link the features with any biological function. In due

consequence, the linking of metabolome and proteome data does not make any biological sense, since the metabolite features are only m/z numbers that cannot be linked to metabolic pathways. Finally, the number of detected proteins is also very moderate since a HPLC-LTQ Orbitrap Velos setup would be able to detect more than 3000 proteins in a cell lysate.

Response: To address the reviewer's concern on our proteomics and metabolomics data, we made further clarifications in following responses as well as appropriate changes in our manuscript.

In terms of the number of detected features by HPLC-MS analysis, the reviewer considered our number (3400-8000) far too high to represent real metabolites detectable in a simple HPLC-MS analysis. First, we would like to point out that the number of detected features is not equal to the real metabolite number. The word "detected features or peaks" is defined by a unique combination of a mass-to-charge ratio (m/z) and retention time (Nature Methods, 2016, 13(9): 770–776), which is a popularly-used idiom in untargeted metabolomics studies to evaluate the diversity of metabolites in samples. Second, we believe that the number of detected features in THP-1 cells is reasonable and comparable to literature data. We searched 62 metabolomics-related papers published by Nature groups in 2017-2018 (Figure R8). Among them, eight papers performed untargeted metabolomics analysis in cell samples using LC-MS or flow injection MS methods, which are very similar to our analytical approach. And four of them reported the number of detected metabolite features at 980-15000 ranges (Table R3). The number of detected features in our study falls into this range.

To address the reviewer's concern on the false positives in metabolomics analysis, we explained the detailed processes for data analysis, compared different methods to eliminate background noises and made appropriate changes in revised manuscript. **First**, we would like to explain the details for metabolomics data analysis. In our study, we used a reported method to process the untargeted metabolomic data (Analytical chemistry, 2015, 87, 884–891; Current Protocols in Bioinformatics, 2016, 55:14.10.1-14.10.91), including four steps: 1) The raw data from LC-MS analysis was converted into mzXML data format by proteoWizard software. 2) The raw data was uploaded to the XCMS platform (<https://metlin.scripps.edu/xcms/>) for peak detection, retention time collection and alignment. The parameters were set according to a previous study by Dr. Gary Siuzdak, who is the developer of XCMS platform in Scripps Research Institute (Analytical chemistry, 2015, 87, 884–891). The detailed information was described in the revised manuscript (lines 622 to 626), as "The parameters were set as follows: centWave settings for feature detection (maximal tolerated m/z deviation = 15 ppm, minimum peak width = 10 s and maximum peak width = 80 s), obiwarp setting for retention time correction (profStep = 1), and mzwid = 0.015, minfrac = 0.5, and bw = 5 for chromatogram alignment." 3) The resulted feature table from XCMS compiled with m/z, retention time and intensity were exported to an Excel spreadsheet for processing. Relative quantification of metabolite

features were performed as following processes: peak intensities were normalized to the internal standards: L-Methionine-(methyl-13C, d3) (m/z, 154.077) for positive mode and D-Glucose-1,2,3,4,5,6,6-d7 (m/z, 186.099) for negative mode. 4) We compared the normalized peak intensities between control and particle-treated samples in MetaboAnalyst 4.0 (<http://www.metaboanalyst.ca>) implemented by R platform for statistical analysis. The parameters for data pro-processing in MetaboAnalyst were described in lines 634-642, as “The parameters for data pro-processing in MetaboAnalyst were set as follows: removing features with >90% missing values and estimating missing values using k-nearest neighbor (KNN) method; using interquartile range (IQR) method for data filtering to remove features from baseline noises; normalization by reference feature (154.077 for ESI+ data and 186.099 for ESI- data), no data transformation and pareto scaling were chosen for normalization procedure. For statistics, non-parametric ANOVA (Kruskal Wallis Test) was performed, and the adjusted p-value (FDR) cutoff is 0.05. A list of features with their p-values and FDR values was downloaded from the software.” **Second**, we compared different data filtering methods to eliminate background noises. We agree with the reviewer that majority of detected features in sample is also present in the background signals of blank. The average peak intensities in bank and samples were displayed in a heatmap (Figure R5). In our study, ca. 90% of the detected features in samples could also be detected in blanks. However, 65% of features in samples are orders-of-magnitude stronger than the peaks in blanks. Actually, to eliminate the false positives, we used an interquartile range (IQR) data filtering method in MetaboAnalyst to remove the background noises. We thus identified 1867 and 938 significant features for ESI+ and ESI- data, respectively. IQR filtering method can remove features with consistently low intensity values and low variance across the samples (Bioinformatics and Computational Biology Solutions Using R and Bioconductor, Springer Publications 2005, 232-233). IQR provides a measure of the spread of the middle 50% of the intensities for each feature. The IQR is defined as the 75th percentile - the 25th percentile. Typically, the IQR filtering will remove 40% of the features based on their IQR ranking. There are 7 kinds of data filtering methods in MetaboAnalyst, including IQR, standard deviation (SD), median absolute deviation (MAD), relative standard deviation (RSD), non-parametric relative standard deviation (NpRSD), mean intensity value (MeanI) and median intensity value (MedI) (Current Protocols in Bioinformatics, 2016, 55:14.10.1-14.10.91). We used all these data filtering methods to process our data and compared the numbers of identified significant features. As shown in Figure R6, there are limited differences among these methods.

We agree with the reviewer that it is important to identify the metabolite features to link with biological function. Recently, Dr. Fraenkel developed a PIUMet platform for putatively identification of peaks detected in untargeted metabolomics and to infer molecular-associated pathways. (Pirhaji et al, Nature methods, 2016, 13(9): 770–776). To address the reviewer’s concern, we used PIUMet for putatively identification of the discriminating features in our study and performed a pathway analysis for the

potential metabolite in the MetaboAnalyst. As shown in Supplementary Excel Data, we identified 417 putative metabolites. The pathway analysis results are shown in Figure R7, which is also included in the revised manuscript as Figure S3. Description and discussion on this result were added in lines 217 to 227, as: “Putative identification of the discriminating features was conducted using a PIUMet platform, which was developed for untargeted metabolomics by Pirhaji *et al*³³. 314 out of the 2805 identified features are matched to 417 potential metabolites in the HMDB database (Supplementary Excel Data: Sheet1). As shown in Figure S3, metabolic pathway analysis with MetaboAnalyst revealed that the putatively identified metabolites were responsible for 14 pathways including sphingolipid metabolism, tryptophan metabolism, phenylalanine metabolism, pyrimidine metabolism, glycerophospholipid metabolism, beta-alanine metabolism, D-glutamine and D-glutamate metabolism, tyrosine metabolism, purine metabolism, pantothenate and CoA biosynthesis, sulfur metabolism, glutathione metabolism, propanoate metabolism and primary bile acid biosynthesis (Supplementary Excel Data: Sheet2)”, lines 456-482, as: “To explore Fe₂O₃-induced bio-effects, metabolomics and proteomics analyses were performed in THP-1 cells exposed to Fe₂O₃ library. Among the 2854 significant metabolite peaks, 417 putative metabolites were identified from 314 features. Metabolic pathway analysis by MetaboAnalyst revealed that the putatively identified metabolites are responsible for 14 metabolic pathways (Figure S3). These pathways were reported to have close relationships with Fe₂O₃-induced proteomics pathway changes (Supplementary Excel Data: Sheet3). The identified metabolic pathways could also reflect the function changes of subcellular organelle including lysosome and mitochondria in Fe₂O₃-treated THP-1 cells. Among the 14 metabolic pathways, sphingolipids metabolism is significantly altered in the Fe₂O₃-treated samples ($p=0.017$). Sphingolipids is bioactive lipids that related to a diverse range of cellular responses, including cell proliferation, autophagy and inflammation⁴³. The regulation mechanisms of various sphingolipids, such as ceramide⁴⁴, sphingosine-1-phosphate⁴⁵, ceramide-1-phosphate⁴⁶ and glycosphingolipids⁴⁷, in inflammatory processes have been extensively studied. Interestingly, while there are 2 ceramides, 2 sphingosine-1-phosphates and 1 glucosylceramide in the list of 417 potential metabolites, proteomics identified a significant changed protein, a single precursor of sphingolipid activator protein (PSAP). This is a soluble membrane protein of lysosome and can be converted to sphingolipid activator protein, which is essential for the hydrolysis of sphingolipids in lysosomes⁴⁸. Besides, sphingosine-1-phosphate was demonstrated to play important roles in regulating cell migration *via* the G-protein-coupled receptors S1P⁴⁹. Phosphatidic acid, an important member in glycerophospholipid metabolism has been demonstrated to be a physiological regulator of ceramide 1-phosphate stimulated macrophage migration⁵⁰. These metabolite pathways show correlations with the identified protein signaling changes, and well support our findings on Fe₂O₃-induced NLRP3 inflammasome activation as well as MCP-1 dependent migration in THP-1 cells.”

In this study, we identified 1699 proteins and 785 of them (proteins that can be

detected in at least two of the three replicates) were selected for statistical analysis. We agree with the reviewer that the number of detected proteins is moderate in our study. However, since protein numbers are determined by the HPLC separation and mass spectrometry detection in proteomics analysis, the protein number (1699) in our analytical system (15 cm 1D separation column, 150 min gradient time and LTQ-Orbitrap detection) is reasonable and comparable to literature reports (1500-1900 identified proteins) where they used similar analytical system to us for proteomics analysis in cells (Kocher et al, Anal Chem. 2011, 83, 2699-2704; Li et al, J. Proteome Res., 2012, 11 (3), pp 1582–1590). In addition, we have to point out that this study is aimed to make new findings on nano-SAR and disclosure cellular mechanisms for a few interesting bio-effects of Fe₂O₃, rather than developing new proteomics technology. From this perspective, we believe this research goal have been achieved by our current analysis method.

Table R3 Untargeted metabolomic studies in cell samples using LC-MS or flow injection MS methods

Sample	Methods	Detected features	Putatively identified metabolites	Structurally identified metabolites	References
E. coli	Flow injection TOF MS	4720(ESI+)	962 metabolites for 777 features	N/A	Nature Method. 2017, DOI:10.1038/nmeth.4103
Bone marrow cells	UPLC-MS	14062(ESI+), 5959(ESI-)	343	N/A	Nature communications 2017, DOI: 10.1038/ncomms15621
CD4 T cells	Flow injection TOF MS	7722 (ESI+)	>3000 metabolites for 632 features	NA	Nature 2017 DOI:10.1038/nature22964
CEM T-ALL cells	UPLC-MS	N/A	N/A	N/A	Nature communications 2017, DOI: 10.1038/s41467-017-00221-3
macrophages	UPLC-MS	N/A	N/A	2	Nature immunology 2017, DOI:10.1038/ni.3796
Stem cells	LC-MS	N/A	207	10	Nature Biomedical Engineering 2017, DOI: 10.1038/s41551-017-0127-4
Oligodendrocyte cells	UPLC-MS	N/A	N/A	22	Beyer et al, Nature chemical biology 2018 DOI: 10.1038/NCHEMBIO.2517
T cells	Flow injection TOF MS	989(ESI-)	5125 metabolites for 989 features	4	Nature communications 2018, DOI:10.1038/s41467-018-04274-w

Figure R5 Heatmap visualizing the intensities of detected features (ESI+)

The average intensities of each feature in 9 sample groups (n=3 for each sample group) and blanks (n=10) were calculated, and then imported to Matlab R2009b for log 10-transformation and generating the heatmap.

Figure R6 Scatter plots of the significant features in ESI+ detection model

The normalized data were submitted into Metaboanalyst, and processed by seven data filtering methods including IQR, SD, MAD, RSD, NpRSD, MeanI and MedI. Each significant feature was described by its retention times and m/z.

Figure R7 Metabolite pathway analysis of 417 putatively identified metabolites

HMDB IDs of the 417 potential metabolites were input into the MetaboAnalyst for metabolite pathway analysis. We identified 14 pathways including sphingolipid metabolism (A), tryptophan metabolism (B), phenylalanine metabolism (C), pyrimidine metabolism (D), glycerophospholipid metabolism (E), beta-alanine metabolism (F), D-glutamine and D-glutamate metabolism (G), tyrosine metabolism (H), purine metabolism (I), pantothenate and CoA biosynthesis (J), sulfur metabolism (K), glutathione metabolism (L), propanoate metabolism (M) and primary bile acid biosynthesis (N). Pathway library for Homo sapiens was selected for the analysis. Hypergeometric test was performed in the over representation analysis, while relative-betweenness centrality algorithm was used in the pathway topology analysis.

6. The release of pro-inflammatory cytokines is also a well-known biological effect of exposure to ENM, see Mahmoud Elsabahy and Karen L. Wooley Cytokines as biomarkers of nanoparticle immunotoxicity, Chem.Soc.Rev., 2013, 42 , 5552.

Response: We carefully examined the review paper by Mahmoud et al. This paper summarized the cytokines induced by ENMs and discussed the possibility of using these cytokines as predictive biomarkers of nanoparticle immunotoxicity. They also discussed some mechanisms of nanoparticle immunotoxicity including oxidative stress, TLR signaling, NF- κ B activation and inflammasome activation. However, several important questions regarding ENM-induced inflammation effects are not mentioned in this review paper, e.g. i) whether Fe₂O₃ can induce NLRP3 inflammasome activation; ii) which properties of Fe₂O₃ are responsible for their inflammatory effects; iii) what's the detailed signaling events involved in Fe₂O₃-induced inflammasome activation. In our study, we demonstrated that Fe₂O₃ can induce NLRP3 inflammasome activation via endocytosis of nanoparticles, lysosome damages and cathepsin B release. Surface reactivity and aspect ratios are responsible for the inflammatory effects of Fe₂O₃ plates and rods, respectively. We believe these findings filled the knowledge gaps on ENM-induced inflammation. Elsabahy et al' paper was cited in the revised manuscript as ref 52.

7. In conclusion, I do see a significant number of conceptual and technological

deficiencies in this manuscript such that I believe it is not suitable for publication in Nature Communications.

Response: We hope our clarifications and the new experimental results have addressed the reviewer's concern.

Specific comments

1) Introduction: contains too many etc.: either name what is meant or leave it out

Response: Thanks, we have corrected this as the reviewer suggested.

2) Page 3, line 77: systemS biology is the commonly used term

Response: Thanks, we use "omics" to replace "system biology".

3) Page 3, line 80: genes are not biomolecules, correct would be nucleic acids

Response: Thanks, we corrected this in the revised manuscript.

4) Page 4, line 115: Brunauer-Emmet-Teller (BET) method (https://en.wikipedia.org/wiki/BET_theory)

Response: Thanks, we corrected this in the revised manuscript.

5) Page 7, line 165: ... metabolites ... were subjected to c18 reversed-phase column
-> ... metabolites were separated by reversed-phase HPLC ...

Response: Thanks, we corrected this in the revised manuscript.

6) Page 7, line 168: high-resolution time-of-flight mass spectrometer

Response: Thanks, we corrected this in the revised manuscript.

7) Figure 2 c: This looks very fancy but does not really give a lot of useful information. As stated above, the linking of identified proteins to unidentified metabolite m/z values does not make any biological sense. From the figure I learn that there may be a positive, negative or null correlation between protein expression and metabolite concentration, which is trivial. Such an investigation only makes sense when the correlating metabolites and the metabolic pathways behind are identified.

Response: Thanks, we have removed figure 2C.

8) Figure 3: 3D bar graphs should be generally avoided; effects would be much better visible in a heat map, showing the correlation between ENM properties and biological effects.

Response: Thanks. As the reviewer suggested, we used a heatmap to show the correlation between ENM properties and biological effects.

9) Figure 4: NALP3 and NLRP3 protein/gene names are used promiscuously, should be unified.

Response: Thanks, we unified the protein/gene name as NLRP3 in the revised manuscript.

10) Methods descriptions are insufficient or erroneous in many respects:

a) Page 22, line 529: I do not believe that method is correctly described. Upon extraction of lysates with methanol, metabolites should be in the supernatant and proteins/DNA in the pellet!

Response: Thanks, we have rewritten this part.

b) Relative protein and metabolite quantification and statistics are not sufficiently described. Also details about protein identification are missing.

Response: We have added more information for relative protein and metabolite quantification in the experimental section, lines 621-642, lines 659-663. Protein identification was described in lines 665-673. Following images show the detailed parameter setting during protein identification by MaxQuant (Figure R8).

Figure R8 Parameter settings during the protein identification by MaxQuant

c) The authors utilize a typical microscale ultrahigh pressure HPLC system to run a nanoscale HPLC column (I suppose 75 MICRO m instead of 75 MILLI m i.d.), How was that accomplished? For the HPLC method the authors refer to another published

paper from the M. Mann group (ref. 43), but in the reference no methodological details are given but reference is made to another publication.

Response: Thanks, we corrected these errors. The i.d. of the trap column and the analytical column is 200 and 75 MICROMETER, respectively. The particle sizes of stationary phase are 5 or 3 MICROMETER.

The analysis was performed on an Accela 600 HPLC system, a microscale ultrahigh pressure HPLC system. The flow rate of the mobile phase in analytical column was controlled at 200 nL/min during LC-MS/MS analysis by using a designed chromatographic separation system including a 6-way valve with 4# hole sealed, a 3-way union, a 4-way union, a trap column and an analytical column (Figure R9). The red lines show the sample loading process where the distributary channel 1 is blocked to allow the sample enrichment in the trap column (flow rate of mobile phase at 5 μ L/min). Black lines show the LC-MS/MS analysis process where distributary channel 2 is blocked and distributary channel 1 is connected with analytical column. The flow rate is controlled at 200 nL/min during the analysis process by tuning the pump pressure.

This nano-LC-MS/MS analysis platform was developed by Dr. Fangjun Wang. The detailed information has been well described in previous publications (Wang et al, Analytical Chemistry, 2007, 6599; Liu et al, Analytical Chemistry, 2013, 2847). We have included these references to replace the old one.

Figure R9 Schematic images of the sample loading and analysis process in LC-MS/MS detection

The chromatographic separation system includes a 6-way valve with 4# hole sealed, a 3-way union, a 4-way union, a trap column and an analytical column. The red lines show the sample loading process where the distributary channel 1 is blocked. Black lines show the LC-MS/MS analysis process where distributary channel 2 is blocked and distributary channel 1 is connected with analytical column.

d) Data availability: availability from the corresponding author upon reasonable?? Request is not meeting modern standards. Data must be freely available in public data repositories.

7. **Response:** Metabolomics and proteomics data have been deposited into MetaboLights (<https://www.ebi.ac.uk/metabolights/MTBLS721>; user name: xiaomingcai1982@hotmail.com; password: MTBLS721) and PRIDE (code: PXD010614; username: reviewer38153@ebi.ac.uk ; password: wzJEYRQP), respectively. Other data are available in a generalist repository, Harvard Dataverse (<https://dataverse.harvard.edu/privateurl.xhtml?token=6f339d25-7b45-48da-a2dd-5f733f8d69ea>).

REVIEWERS' COMMENTS:

Reviewer #1 (Remarks to the Author): The authors have made proper revisions.

Editorial note: Reviewer #2 was unable to review the manuscript so a third reviewer (reviewer #3) was invited to address the authors response to reviewer #2 comments.

Editorial note: Reviewer #3 in comments to the editor thought the authors had done a thorough job in addressing all the technical concerns of reviewer #2.